# Structural and functional characterization of *Tg*GSK3, a druggable kinase in *Toxoplasma gondii*

Silvia Diaz-Martin [1,6], Christopher Swale [1,6], Valeria Bellini[1], Irina Dobrescu [2], Janine Wenker[2], Marie-Pierre Brenier-Pinchart [1], Laurence Braun[1], Alwéna Tollec[3], Charlotte Corrao[1], Yohann Couté [3], Caroline Mas [4], Fabrice Laurent [2], Matthew Bowler [5], Mohamed-Ali Hakimi [1] ✉ & Alexandre Bougdour [1] ✉

*Toxoplasma gondii* and *Cryptosporidium* species are apicomplexan parasites of significant medical and veterinary importance. Although current therapeutic options for toxoplasmosis and cryptosporidiosis demonstrate notable efficacy, their clinical efficacy is often limited by suboptimal efficacy and frequent adverse effects. Moreover, therapeutic alternatives remain limited or non-existent, particularly for cryptosporidiosis, for which nitazoxanide is currently the only approved medication to treat diarrhea in adults and children older than 1 year of age. To identify alternative therapeutic options for addressing these health challenges, we performed a phenotypic screening of an FDA-approved drug repurposing library against *Toxoplasma*. This screening identifies LY2090314 as a potent inhibitor of *T. gondii* and *Cryptosporidium* growth in mammalian cells. Through a target deconvolution strategy combining forward genetics, transcriptome sequencing, and computational mutation analysis, we elucidate the parasiticidal mechanism of LY2090314 and demonstrate that *Tg*GSK3 kinase is its primary molecular target. We also report the first X-ray crystal structure of LY2090314 bound to *Tg*GSK3, resolved at 2.1 Å, which reveals an interaction mode characteristic of type I ATP-competitive inhibitors. Furthermore, interactome analysis uncovers functional connections between *Tg*GSK3 and key cytoskeletal and signaling regulators, providing insights into compound's effects. Collectively, these findings validate *Tg*GSK3 as a promising therapeutic target for toxoplasmosis and offer mechanistic insights into apicomplexan GSK3 biology.

Infectious diseases caused by intracellular single-celled parasites of the Apicomplexa phylum, such as *Plasmodium*, *Toxoplasma*, and *Cryptosporidium*, pose significant public health challenges worldwide[1]. These parasites, responsible for malaria, toxoplasmosis, and cryptosporidiosis, respectively, affect millions of individuals worldwide, causing significant morbidity and mortality, particularly in low-income countries, emphasizing the urgent need for effective treatments.

Malaria affects over 240 million people worldwide, with an estimated 608,000 deaths reported in 2022 (World Health Organization, World Malaria Report 2023[2]). Toxoplasmosis is a widespread zoonotic infection, with nearly a third of the world's population being

seropositive[3]. While acute primary infection is often self-limiting and typically resolves without specific treatment in healthy adults, a subclinical infection persists for life in infected individuals, constituting the chronic phase of toxoplasmosis. In the absence of sustained immunity, reactivation of latent forms of *T. gondii* can lead to severe, potentially life-threatening disease, particularly in immunocompromised individuals such as AIDS patients, transplant recipients, and those undergoing chemotherapy[4]. Severe cases may also arise during congenital transmission of the parasite to an unborn child. Additionally, life-threatening acute primary infections have been reported in healthy adults, particularly in regions of South America and Africa[5–10]. These cases are potentially linked to mouse-virulent atypical strains circulating in these regions, which may exhibit greater virulence in humans compared to the canonical strains from Middle East, Europe, and North America[11]. Such strain-specific differences likely contribute to the variability in clinical severity in humans observed across different regions[11–13].

While current medications are effective against Apicomplexa-mediated diseases, the persistent threat of drug resistance and adverse side effects, coupled with limited alternatives, underscores the critical need for the development of new, safe, and economical classes of small-molecule drugs. The current standard treatment for toxoplasmosis, a combination of pyrimethamine-sulfadiazine, is often associated with severe side effects, particularly in immunocompromised individuals[14]. Similarly, artemisinin-based combination therapies remain the cornerstone for malaria treatment; but the emergence and spread of resistance threaten their efficacy. Treatment options for cryptosporidial infections are even more restricted, relying primarily on nitazoxanide. The growing prevalence of drug-resistant parasites underscores the need to exploit the vulnerabilities of human parasites for therapeutic interventions by identifying novel druggable enzyme families.

In this study, we report the identification of the compound LY2090314 from a phenotypic screening of an FDA-approved drug repurposing library as an inhibitor of *T. gondii* growth in human cells. LY2090314, a maleimide-based kinase inhibitor originally developed to target human glycogen synthase kinase-3 (GSK3), has advanced through phase 1/2 clinical trials for treating advanced solid tumors[15] and acute leukemia[16,17]. Here, we present evidence that LY2090314 effectively inhibits *T. gondii* tachyzoite growth in vitro at sub-micromolar concentrations. Using a forward genetic approach based on transcriptome sequencing, we identified the *T. gondii* GSK3 (*Tg*GSK3), a serine/threonine kinase, as the primary enzymatic target of LY2090314. Using an integrated structural biology approach, we uncover the inhibitory mechanism of LY2090314 on the enzymatic activity of the fitness-conferring *Tg*GSK3 and provide a rationale for the resistance mechanisms conferred by the amino acid substitutions identified in the resistant parasite lines. Notably, we present the first X-ray crystal structure of LY2090314 bound to *Tg*GSK3, revealing its mode of interaction characteristic of type I, ATP-competitive inhibitors. Genetic and biochemical evidence further indicate that *Tg*GSK3 functions as a dimer. Overall, this study uncovers a novel and promising drug-target pair that could serve as the foundation for leveraging drug development against apicomplexan parasites.

## Results

### Phenotypic screening identifies LY2090314 as a potent inhibitor of *T. gondii* growth in vitro

In an effort to identify novel drug candidates against *T. gondii* parasites, the compound LY2090314 (compound #209) was identified, alongside altiratinib (compound #246[18],), in a phenotypic screening of 514 approved molecules (Fig. 1a-b). This compound library, originally obtained from TargetMol, comprises FDA-approved drugs and advanced clinical candidates with known safety profiles and mechanistic diversity. It includes inhibitors targeting a wide array of druggable

pathways such as kinases, ion channels, and metabolic enzymes, which may partly explain the relatively high hit rate observed in the primary screen. All hits identified in this screen were systematically followed up, with prioritization based on key criteria: (i) selective inhibition of parasite growth with minimal toxicity to host cells, (ii) activity against multiple apicomplexan species such as *Cryptosporidium parvum*, and (iii) feasibility of target deconvolution. In this context, LY2090314 was prioritized due to its potent and selective activity against *T. gondii*, nanomolar $EC_{50}$ values, favorable selectivity indices, and the successful identification of resistance-conferring mutations that enabled in-depth functional and structural characterization of its molecular target (see below).

LY2090314 demonstrated selective inhibition of tachyzoite growth without affecting the human primary fibroblasts (HFFs) monolayer, as determined by Hoechst staining (Fig. 1a). Its effectiveness was validated at sub-micromolar concentrations, with measured half maximum effective concentration ($EC_{50}$) of 382 nM, which is comparable to pyrimethamine, the standard of care for toxoplasmosis (Fig. 1c and Supplementary Fig. 1a). Immunofluorescence assays using anti-GAP45 antibodies revealed a reduced intracellular parasite growth in LY2090314-treated parasites, accompanied by abnormal cell division (Fig. 1d). Remarkably, complete and sustained inhibition of growth was achieved at 600 nM LY2090314, as parasites were unable to grow and form plaques over the course of seven days (Fig. 1e). LY2090314 also showed low cytotoxicity on host cells (Fig. 1c and Supplementary Fig. 1b-c), particularly with the CellTox Green assay that measures changes in membrane integrity, resulting in a selectivity index of 892 and 808 for HFFs and the non-transformed ARPE-19 human retinal pigment epithelial cell line, respectively (Fig. 1f).

To evaluate whether LY2090314 also affects the chronic stage of the parasite, we assessed its activity against in vitro–induced bradyzoites using the ME49 pGRA1-dsRed2.0 pBAG1-mNeonGreen dual-reporter strain. Under bradyzoite-inducing conditions, treatment with 600 nM LY2090314 for 72 h resulted in a marked reduction in expression of the bradyzoite-specific markers BAG1, BCLA, and BSM[19,20], as measured by fluorescence intensity and immunostaining (Supplementary Fig. 1f–k). In contrast, expression of the tachyzoite-associated GRA1 reporter (dsRed2.0) was increased, indicating a shift in gene expression dynamics. These observations suggest that LY2090314 exerts a biological effect on in vitro bradyzoites, in agreement with the documented expression of *Tg*GSK3 in this life stage (ToxoDB.org). In comparison, pyrimethamine treatment produced only modest changes in bradyzoite marker expression or GRA1 levels, indicating that LY2090314 may act through a distinct and potentially more specific mechanism to modulate developmental gene expression in *T. gondii*.

In addition to its efficacy against *T. gondii*, LY2090314 demonstrated potent efficacy in inhibiting the growth of *Cryptosporidium* parasites, with $EC_{50}$ values even lower than those observed for *T. gondii*, reflecting a higher antiparasitic potency (Fig. 1g and Supplementary Fig. 1d-e). Notably, these $EC_{50}$ values were also lower than those obtained for indirubin E804, another potential GSK3 inhibitor tested in parallel, further highlighting the superior efficacy of LY2090314. Indirubin E804 had been previously identified in a crystal structure of *Cp*GSK3 (cgd4_240) in complex with the inhibitor (PDB ID: 3EB0). Note that no toxicity was detected up to 100 μM on HCT-8 human host cells (Fig. 1f), suggesting SI of a least 350 and 625 for *C. parvum* INRAE and IOWA strains, respectively (Fig. 1h).

To assess whether this inhibitory profile was specific to LY2090314, we examined several additional GSK3 inhibitors present in the original compound library. These included chemically diverse scaffolds such as aminopyrimidines, imidazopyridines, and thiadiazolidinones. None of these compounds significantly impaired *T. gondii* growth in vitro, suggesting limited efficacy or insufficient engagement with the parasite kinase (Fig. 1a). To further investigate the potential

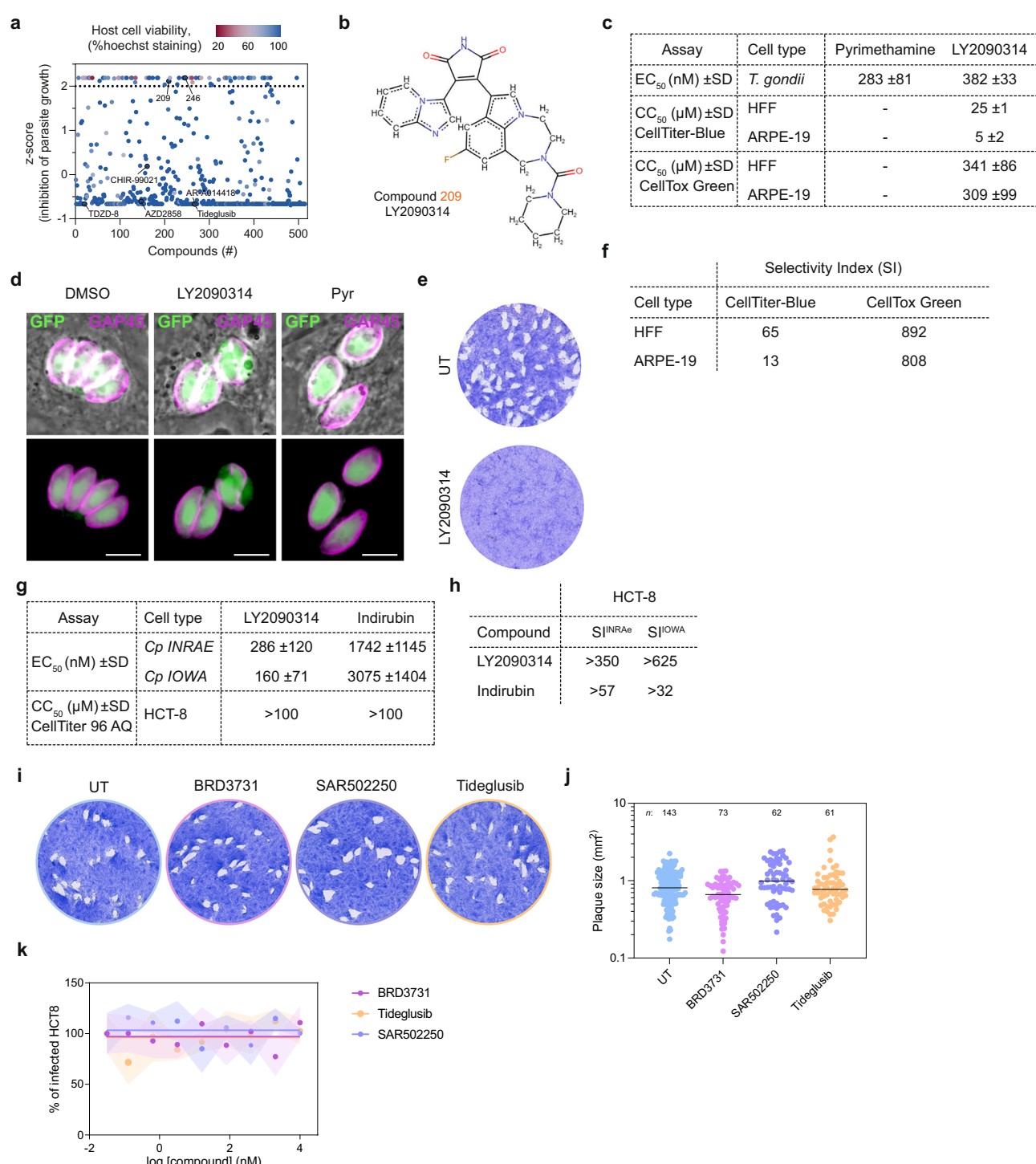

for GSK3 inhibition, we evaluated three additional GSK3 inhibitors with reported favorable safety profiles - BRD3731, SAR502250, and Tideglusib[21]. None of these compounds inhibited the proliferation of *T. gondii* or *Cryptosporidium* under the same experimental conditions (Fig. 1i-k), further underscoring the unique antiparasitic activity and specificity of LY2090314 against apicomplexan parasites.

### Forward genetic screening identifies *Tg*GSK3 as the enzyme targeted by LY2090314

To gain insights into the mechanism of action of LY2090314 in *T. gondii*, we used a previously developed workflow based on a forward genetic screen combined to RNA-Seq analysis, variant calling, and computational mutation discovery[22]. Central to this approach is that

the gene mutated in more than one independently mutagenized resistant clone might be relevant to the mechanism of drug resistance, thereby alleviating the notoriously difficult molecular mapping of point mutations induced by mutagenic agents. First, eight independent ethyl methane sulphonate (EMS) mutagenesis experiments were performed, and the resulting mutagenized parasites were selected in the presence of 600 nM LY2090314 (Fig. 2a), which corresponds to ~1.6 times the EC$_{50}$ value. Resistant parasites emerged ~10 days later and were cloned by limited dilution (Supplementary Fig. 2a). A single clone from each mutagenesis experiment, designated A to H, was analyzed by whole-genome RNA-seq (Fig. 2a-d). To map the EMS-induced mutations conferring resistance against LY2090314, Illumina sequencing reads were aligned to the *T. gondii* GT1 reference genome.

**Fig. 1 | Effectiveness of LY2090314 against apicomplexan parasites and human host cells. a** Distribution of z-scores for each compound, calculated from the percentage of growth inhibition of *T. gondii* tachyzoites. The dotted line represents the cut-off mean z-score, set to select compounds with a composite z-score greater than 2, corresponding to at least 90% inhibition of parasite growth. Hoechst staining was used as a proxy for host cell viability; points are colored according to the percentage of Hoechst signal. Compounds LY2090314 (209) and the previously characterized inhibitor altiratinib (246) are highlighted. Additional GSK3 inhibitors included in the compound library are also indicated. **b** Chemical structure of the maleimide-based kinase inhibitor LY2090314. **c** Table showing the half-maximal effective concentration (EC$_{50}$) values for pyrimethamine and LY2090314 against *T. gondii* parasites. EC$_{50}$ data are presented as means ± SD from three independent biological replicates, each with three technical replicates. The half-maximal cell cytotoxicity concentration (CC$_{50}$) values for LY2090314 were determined using the CellTiter-Blue or CellTox Green assay kits on HFF and ARPE-19 human cells. CC$_{50}$ data are presented as means ± SD from at least two independent biological replicates, each with three technical replicates. Associated data are shown in Supplementary Fig. 1a–c. **d** Immunofluorescence microscopy showing the effects of LY2090314 on intracellular *T. gondii* parasites within HFFs. HFF cells were infected with tachyzoites (RH NLuc) and incubated with 600 nM LY2090314, 1 μM

pyrimethamine (Pyr), or 0.1% DMSO as control. Cells were fixed 24 h post-infection and stained with antibodies against the *T. gondii* inner membrane complex protein GAP45 (magenta). Cytosolic GFP is shown in green. Scale bars, 5 μm. **e** Plaque assays showing the restrictive effect of LY2090314 on *T. gondii* lytic cycle. **f** The selectivity index (SI) for *T. gondii* parasites was calculated as the ratio of the human CC$_{50}$ to the parasite EC$_{50}$. **g** Table showing the EC$_{50}$ values for LY2090314 and indirubin against *C. parvum* parasites. Mean EC$_{50}$ values ± SD from at least three independent biological replicates are shown. Associated data are shown in Supplementary Fig. 1d, e. The CC$_{50}$ data were determined using the MTS assay on HCT-8 human host cells. (*n* = 2). **h** SI for *C. parvum* INRAE and IOWA strains are indicated. **i** Evaluation of the GSK3 inhibitors BRD3731, SAR502250, and Tideglusib in plaque assays showing their lack of efficacy against *T. gondii* tachyzoites. **j** Quantification of plaque sizes shown in (**i**). The number of plaques analyzed is indicated above the figures. **k** Evaluation of BRD3731, SAR502250, and Tideglusib on *C. parvum* INRAE-nLuc-mCherry strain. Dose–response curves showing the percentage of host cells infected by *C. parvum* in response to increasing concentrations of the indicated compounds. Data were fitted using non-linear regression analysis. Results are presented as mean ± SD, with shaded error envelopes, based on at least six replicate measurements. Source data are provided as a Source Data file.

The assembled sequences were analyzed to identify single nucleotide variations (SNVs), small insertions, or short deletions using the parental strain as a reference, (see Methods section and Supplementary Data 2). By focusing on mutations in coding sequences, we identified a single gene, *TGGT1_265330*, which harbored SNVs leading to amino acid substitutions in each of the eight drug-resistant lines, absent in the parental strain. These mutations resulted in amino acid substitutions at the protein residues G54, H129, and T133 (Fig. 2e and Table 1). *TGGT1_265330* encodes for a putative cell-cycle-associated protein kinase of 394 amino acids (Fig. 2e). Notably, each resistant clone harbored only a single point mutation in the *Tg*GSK3 coding sequence, all identified mutations are confined to the serine/threonine protein kinase catalytic domain: G54D, affecting the first residue of the glycine triad, and H129D and T133M, both positioned in the hinge region preceding the C-lobe (Fig. 2e). The T133M mutation stands out as the most frequently observed, detected in six out of the eight sequenced clones. Bulk nanopore DNA sequencing analysis of the *TGGT1_265330* alleles in the selected resistant parasite population exclusively identified the T133M mutation (Supplementary Fig. 3b). This observation suggests that the T133M mutation is either more prone to being induced by the EMS alkylating agent or provides a significant selective advantage compared to other mutations.

Phylogenetic analysis identified TGGT1_265330 as a member of the GSK3 clade, with conservation observed in other apicomplexan parasites, including *Plasmodium falciparum* and *Cryptosporidium parvum* (Fig. 2f and Supplementary Fig. 2b). This finding aligns with the observed inhibitory effect of LY2090314 on *C. parvum* growth in vitro (Fig. 1g and Supplementary Fig. 1d–e). Consequently, TGGT1_265330 is hereafter referred to as *Tg*GSK3. Immunofluorescence analysis of intracellularly growing parasites revealed a nucleo-cytoplasmic localization of *Tg*GSK3-HA (Fig. 3g; Supplementary Fig. 3c, d). *Tg*GSK3 was identified as a fitness-conferring gene for tachyzoites, as its genetic disruption results in a fitness score of −4.12[23]. Conditional knock-out of this kinase using the DiCre/loxP system[24] confirmed its essential role in tachyzoite growth (Fig. 3i and j), with complete protein depletion observed 48 h after Rapamycin treatment (Fig. 3g and h; Supplementary Fig. 3e and f).

To verify that the mutations in *TGGT1_265330* were sufficient to confer resistance to LY2090314, we reintroduced each mutation identified in LY2090314-resistant parasites into the susceptible parental wild-type strain. This was achieved using the CRISPR/Cas9 system in combination with homology-directed repair for gene editing in *T. gondii* (Fig. 3a). After selection with LY2090314, emerging resistant parasites were cloned (Fig. 3b and c), and DNA sequencing confirmed

that the mutations were correctly inserted into *TGGT1_265330* (Supplementary Fig. 3a). In genome-edited parasites, the *TGGT1_265330* point mutations G54D, H129D, and T133M conferred reduced susceptibility to LY2090314 compared to wild-type parasites, with EC$_{50}$ values increasing by ~2.9-fold, 4.5-fold, and 4.8-fold, respectively (Fig. 3b-e). CRISPR/Cas9−edited parasites carrying the G54D mutation exhibited lower resistance to LY2090314 compared to those harboring the H129D or T133M mutations. This observation aligns with results from EMS-mutagenized parasites and further suggests that the T133M or H129D mutations confer a higher level of resistance to the compound compared to the G54D mutation (Figs. 2b-c and 3b-e). Collectively, these data support the hypothesis that LY2090314 interferes with *Tg*GSK3 activity, resulting in impaired parasite growth.

## Structural insights into the mechanism of action of LY2090314 on *Tg*GSK3 and resistance-conferring mutations

To investigate the molecular mechanism underlying LY2090314 inhibition, we produced the kinase domain of *Tg*GSK3, excluding the intrinsically disordered N-terminal region, which was anticipated to hinder crystallization (Supplementary Fig. 4a). Using baculovirus-infected insect cells, two N-terminal truncations (starting at the residues 20 or 46) were successfully expressed and further purified. Successful co-crystallization conditions were achieved by incubating purified *Tg*GSK3 with LY2090314 prior to final size exclusion chromatography and concentration. From these two constructs, two crystal types were obtained (statistics are shown on Supplementary Table 1): an orthorhombic system diffracting to 2.1 Å with one copy of *Tg*GSK3 per asymmetric unit, and a monoclinic system diffracting at 3.0 Å with a crystallographic tetramer of *Tg*GSK3 per asymmetric unit. Although LY2090314 density was clearly visible in both derived structures and in all copies of *Tg*GSK3 within the monoclinic system, structural representations are derived from the orthorhombic crystal structure. *Tg*GSK3 adopts the characteristic kinase fold, divided into two sub-domains classified as the N- and C-Lobe (Fig. 4a). Overall, *Tg*GSK3 has high sequence (54% identity) and structural conservation (r.m.s.d. of 1.67 Å) to the human ortholog (Supplementary Fig. 4b), with most of the structural variation located within loops in the N-lobe (N-terminal region, Supplementary Fig. 2b). The LY2090314 compound is positioned within the ATP binding pocket, stabilizing phenylalanine 196, a key residue within the DFG motif, in an allosteric pocket bound or "DFG-in" conformation.

LY2090314 is therefore classified as a "type I" inhibitor, competing with ATP for binding in the ATP binding site. The high-resolution

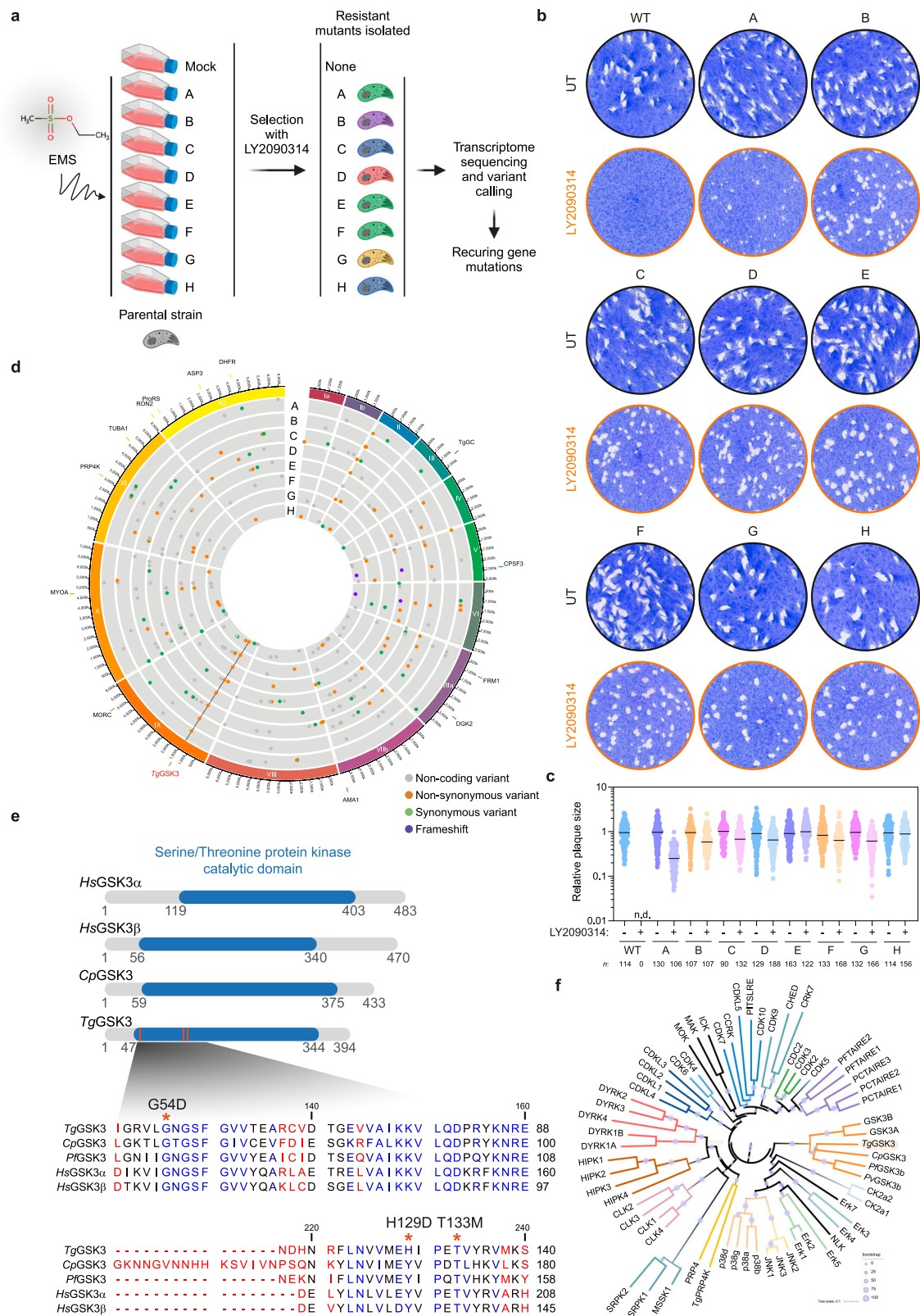

electron density allowed for accurate modeling of the intricate folding of LY2090314 within the binding cavity (Supplementary Fig. 4c). Similar to other maleimide-based kinase inhibitors (Supplementary Fig. 4e and f), the maleimide group interacts with the hinge region of *Tg*GSK3 (Fig. 4b), forming hydrogen bonds with the backbone residues E128 and I130 (Fig. 4b; Supplementary Fig 4d). In the vicinity of these direct ligand to protein bonds, LY2090314 also interacts indirectly with *Tg*GSK3 by coordinating 3 water molecules within the cavity

(Fig. 4b and c; Supplementary Fig. 4d). One of these water molecules forms a tetragonal coordination with both LY2090314 and 3 side chains of *Tg*GSK3, including the catalytic lysine K76 (Fig. 4b and c; Supplementary Fig. 4d). It is worth noting that homology-based docking of LY2090314 would have resulted in a partially incorrect binding mode hypothesis. Indeed, the closest homologous compound, M97 (which binds to C-met and features a quinolinium side moiety instead of the imidazopyridine), is flipped in this perspective by 180°

**Fig. 2 | Target identification for LY2090314 in *T. gondii* parasites. a** EMS-based forward genetic strategy to isolate LY2090314-resistant *T. gondii* parasites. From each mutagenesis experiment, a single clone (A to H) was isolated and analyzed by RNA-seq. Created in BioRender. Hakimi, M. (https://BioRender.com/p5820hv) (**b**) LY2090314-resistant parasites form plaques after 7 days of growth in the presence of 600 nM LY2090314. **c** Quantification of plaque sizes of wild-type parasites and resistant lines (A to H) when cultured in the presence or absence of LY2090314. n.d., not detected. Data are mean from $n = 3$ independent experiments. **d** Circos plot summarizing single nucleotide variants (SNVs), insertions, and deletions detected by transcriptomic analysis of the *T. gondii* LY2090314-resistant lines, grouped by chromosome (numbered in Roman numerals with size intervals given outside). Each dot in the eight innermost gray tracks corresponds to a scatterplot of the mutations identified in the coding regions of the seven drug-resistant strains, with each ring representing one of the eight drug-resistant lines (A–H). Each bar in the outermost track represents locations of selected archetypal essential genes. See Supplementary Data 2 for transcriptomic analysis. **e** Schematic representation of the GSK3 protein architecture across different GSK3 isoforms from various species. The schematic at the top displays the conserved GSK3 catalytic domains (blue)

within the GSK3 homologs from *Homo sapiens Hs, Cryptosporidium parvum Cp, Plasmodium falciparum Pf,* and *Toxoplasma gondii Tg*. Positioning of residues that were mutated in parasites resistant to LY2090314 (G54D, H129D, and T133M) are indicated in the *Tg*GSK3 catalytic domain as orange bars. The zoomed sequence alignment below shows the amino acid sequences of the catalytic domains, with conserved residues highlighted in blue, while divergent amino acids are in red. The positions of mutations in *Tg*GSK3 are marked with asterisks. The alignment was generated with CLC Sequence Viewer. **f** Phylogenetic tree of CMGC protein kinases. The tree illustrates the evolutionary relationships among various protein kinases, with branches color-coded to represent different kinase families. Bootstrap values are indicated by circle sizes. The tree was computed using the neighbor-joining algorithm implemented in Clustal Omega, with four combined iterations and the *dealign input* option enabled. The analysis was based on a hidden Markov model (HMM)-guided multiple sequence alignment, which included full-length sequences from the human kinome repertoire, along with GSK3 protein sequences from *Cryptosporidium parvum Cp, Toxoplasma gondii Tg, Plasmodium falciparum Pf, and Plasmodium vivax Pv*. The resulting tree was visualized and annotated using iTOL (Interactive Tree Of Life). Source data are provided as a Source Data file.

on the central maleimide moiety in this model (Supplementary Fig. 4e and f) while maintaining the same central maleimide hydrogen bonds. At the opposite side of LY2090314, the only direct hydrogen bond with an amino acid side chain involves the previously identified threonine 133 (Fig. 4b; Supplementary Fig. 4d), underscoring the importance of this residue in LY2090314 binding. To further investigate how these interactions are affected by the resistance conferring mutations, we used the recombinantly expressed *Tg*GSK3 to assess the consequences of the three identified mutations (G54D, H129D and T133M) on compound binding and kinase activity rescue, comparing these results to the recombinant wild-type *Tg*GSK3 (Fig. 4d and e).

To determine binding affinity, we used microscale thermophoresis to measure the apparent $K_d$ ($K_D$) of LY2090314 against *Tg*GSK3, which was found to be in the low nanomolar range at 34 nM (Fig. 4d). Similarly, the apparent $IC_{50}$ of *Tg*GSK3 was estimated at 112 nM when measured using the kinase-Glo© endpoint ATP consumption assay in the presence of a *Hs*GSK3 substrate peptide (Fig. 4e). The G54D only confers a small $K_D$ and $IC_{50}$ increase (1.8 and 1.7-fold, respectively, Fig. 4e), which can be structurally explained by the location of this residue on the glycine-rich loop of *Tg*GSK3 (Fig. 4b, Supplementary Fig. 5a). This loop folds to accommodate the fluorine moiety of LY209314, forming hydrophobic contacts with it through the C-α of the G54 residue. The mutation to an aspartate side chain does not directly affect the binding (Supplementary Fig. 5a), but rather induces changes in the φ and ψ dihedral angles, likely causing the slight reduction in apparent affinity. Similarly, the H129D mutation results in modest $IC_{50}$ increases (2.2-fold, Fig. 4e) but causes a more pronounced decrease in affinity towards *Tg*GSK3 (6.7-fold, Fig. 4d). Again, the structural mechanism of resistance relies on a similar basis as the histidine side chain is oriented outside and does not directly interact with LY209314 (Supplementary Fig. 5b). Similarly to G54D, H129D also operates more indirectly by perturbing the upstream and downstream C-α dihedrals which are forming direct H-bonds with the inhibitor. Finally, the T133M mutation confers the most dramatic increase in $K_D$ (18-fold, Fig. 4d) and $IC_{50}$ (5.3-fold, Fig. 4e). This is clearly explained structurally as the T to M side chain mutation will abolish one of the key H-bond interactions and at the same time introduce a significant steric clash to LY209314 binding (Supplementary Fig. 5c).

### Homodimerization and redox regulation of *Tg*GSK3: insights into its activation mechanism

Deciphering a protein's biological function often hinges on uncovering its interacting partners. To probe the role and regulation of *Tg*GSK3, we performed in vivo biotinylation using the proximity labeling approach employing BioID2, with *Tg*GSK3 as the bait protein (Fig. 5a and b). The endogenously tagged *Tg*GSK3-BioID2-HA fusion protein

showed a nucleo-cytoplasmic distribution, consistent with the localization observed for the HA-tagged protein alone (Fig. 3g). The biotinylation activity of the *Tg*GSK3-BioID2-HA fusion protein was validated using fluorophore-conjugated streptavidin. In the absence of exogenous biotin supplementation, the primary biotinylation signal was restricted to the apicoplast, reflecting the activity of metabolic enzymes that utilize biotin as a co-factor, as previously reported[25]. However, with biotin supplementation, streptavidin signals were detected in the nucleus and cytoplasm, alongside a pronounced signal at the posterior pole of intracellular tachyzoites. These observations confirmed the biotinylation activity of the *Tg*GSK3-BioID2-HA fusion protein and indicated that specific proteins in these subcellular compartments were labeled (Fig. 5a). To identify potential interacting partners or substrates of *Tg*GSK3, biotinylated proteins were enriched from lysates using streptavidin affinity purification. As a control, a mock experiment was performed using parasites lacking the BioID2 fusion protein (RH *ku80*). Mass spectrometry-based proteomic analysis identified ~200 proteins with ≥5-fold enrichment in abundances in the two biological replicates obtained from *Tg*GSK3-BioID2-HA expressing parasites compared to the mock control (Fig. 5b; Supplementary Data 3). Gene Ontology (GO) enrichment analysis revealed that these candidate interactors were associated with diverse molecular functions, including RNA binding (Fig. 5d). Notably, *Tg*GSK3 was highly enriched ( > 60-fold) in the proximity labeling dataset, likely due to intramolecular biotinylation.

To further explore the *Tg*GSK3 interactome, a yeast two-hybrid (Y2H) screen was performed using a *T. gondii* cDNA library as prey. The bait construct consisted of the catalytically inactive N-terminal *Tg*GSK3^K76H mutant (amino acids 1–394) fused to the LexA DNA-binding domain (*Tg*GSK3^K76H-LexA, Fig. 5c). This kinase-dead *Tg*GSK3 mutant was intended to increase the likelihood of capturing substrate interactions[26]. From a total of 47.4 million cDNA clones screened, 246 positive hits corresponding to 44 unique proteins were identified. These interactors were ranked using a Global Predicted Biological Score (Global PBS) from A to D, with 'A' representing the highest confidence of interaction (Fig. 5e, Supplementary Data 4). Notably, *Tg*GSK3 was classified as an 'A' category interactor, suggesting *Tg*GSK3 self-association. Additionally, interactions between *Tg*GSK3 and the kinase SRPK, the Apical Cap protein AC8, as well as two hypothetical proteins (TGME49_228150 and TGME49_214090), already identified in proximity labeling experiments, were confirmed through this integrative analysis (compare Fig. 5b and e). To further assess the plausibility of direct physical interactions, we performed AlphaFold-Multimer predictions using AlphaFold3[27] for all candidate protein pairs identified by both BioID2 and Y2H. While this approach supported several candidate interactions (Supplementary Data 5), it did

**Table 1 | Mutations found in candidate gene identified by whole-transcriptome analysis by RNA-Sequencing**

| | | | | Variant calling | | | | | | | | |
| | | | | Parental strain | Resistant mutants | | | | | | | |
| Chr. | Gene | Annotation | Position | WT | A | B | C | D | E | F | G | H |
|---|---|---|---|---|---|---|---|---|---|---|---|---|
| IX | TGGT1_265330 | TgGSK3 | 1431960 | | | T133M (ACG to ATG) | T133M (ACG to ATG) | T133M (ACG to ATG) | T133M (ACG to ATG) | T133M (ACG to ATG) | | T133M (ACG to ATG) |
| IX | TGGT1_265330 | TgGSK3 | 1431973 | | | | | | | | H129D (CAC to GAC) | |
| IX | TGGT1_265330 | TgGSK3 | 1432329 | | G54D (GGC to GAC) | | | | | | | |

Gene names are italicized.

not detect high-confidence structural models for interactions between TgGSK3 and SRPK, AC8, TGME49_228150, or TGME49_214090. This limitation may reflect the transient or low-affinity nature of these associations, particularly in the case of kinase-substrate interactions, which are often difficult to capture with current structural prediction tools. Importantly, the majority of the identified TgGSK3's interactors are indispensable for tachyzoite proliferation (Fig. 5f), as defined by genome-wide CRISPR-based fitness screens[23]. This observation is consistent with the essential role of TgGSK3 in parasite viability and supports its functional integration into key signaling and regulatory networks in T. gondii.

The kinase-dead Y2H screen identified TgGSK3 as its own primary interaction hub, suggesting a mechanism of homo-dimerization or multimerization. This self-interacting ability has also been observed in the metazoan homolog GSK3-β, where it plays a pivotal role in activating various serine threonine kinases[28]. In solution, however, the recombinant TgGSK3 expressed in insect cells predominantly exists as a monomer (Supplementary Fig. 6a), indicating that homo-oligomerization process does not occur spontaneously, but is instead dependent on external factors such as substrate binding, co-factors, interacting partners, or post-translational modifications. In this regard, during crystallization trials of TgGSK3 with LY2090314, an intriguing observation was made: alongside the standard crystal forms, a monoclinic crystal lattice emerged after several weeks, diffracting at lower resolution (3.0 Å) and containing four TgGSK3 subunits within the asymmetric unit. These four subunits can be reduced to two homodimers, interacting through non-crystallographic symmetry. In each dimer, TgGSK3 forms a face-to-face interaction (Fig. 5g), relying partly on the seed region identified in the yeast 2-hybrid screening, which spans most of the N-lobe. Additionally, the dimer is crosslinked by a disulfide bridge formed between the cysteine 213 of adjacent monomers (Fig. 5g; Supplementary Fig. 6B). This peculiar cysteine, located on the activation loop just two residues downstream of the phosphotyrosine 211, requires significant structural rearrangements to enable bond formation with the neighboring subunit when compared to the monomeric orthorhombic crystal structure of TgGSK3 (Supplementary Fig. 6b and c). Despite these structural rearrangements, the activation loop remains in a DFG "in" conformation due to the interaction with LY2090314. This trans-disulfide bridge could initially be considered a crystallization artifact. However, cysteine-mediated crosslinked kinase dimers are rare, with only a handful of crystal structures in the PDB available for comparison. Representative examples include PDB IDs: 1ZMW (MARK2, CAMK family), 2WNT (ribosomal protein S6 kinase, RSK family), 5NG0 (RIP2K, Tyrosine Kinase-Like [TKL] superfamily), 4CZT (CIPK23 from *Arabidopsis thaliana*, CBL-interacting serine/threonine-protein kinase), 4EQU (STK10, STE family), and 6VPI and 6VPJ (Aurora kinase A, part of the AGC-related family). Notably, the human Aurora kinase (6VPJ) stands out, as its disulfide-linked cysteine (C290) occupied an equivalent position as the cysteine forming the disulfide bond in TgGSK3 (Supplementary Fig. 6f and g), despite the low sequence conservation in the activation loop region (Supplementary Fig. 6e). Similarly, this cysteine is found two residues downstream of the activation loop phosphotyrosine (Y211) in GSK3 and two residues after a phosphothreonine in Aurora kinase (T288, Supplementary Fig. 6e). In their study, the authors demonstrated that the conserved cysteine residue (C290 in Aurora) plays a critical role in autophosphorylation, driving kinase pathway activation and potentially linking this process to redox state transitions during mitosis[29]. While these findings may not directly translate to TgGSK3, given the distinct functional role of the two kinases, they highlight a broader role for this cysteine in regulating dimerization and activation through redox signaling. Together, the Y2H results and the monoclinic crystal structure suggest that the activation pathway of TgGSK3 in T. gondii relies on a homodimerization process, likely controlled by the conserved cysteine 213, a feature preserved across species.

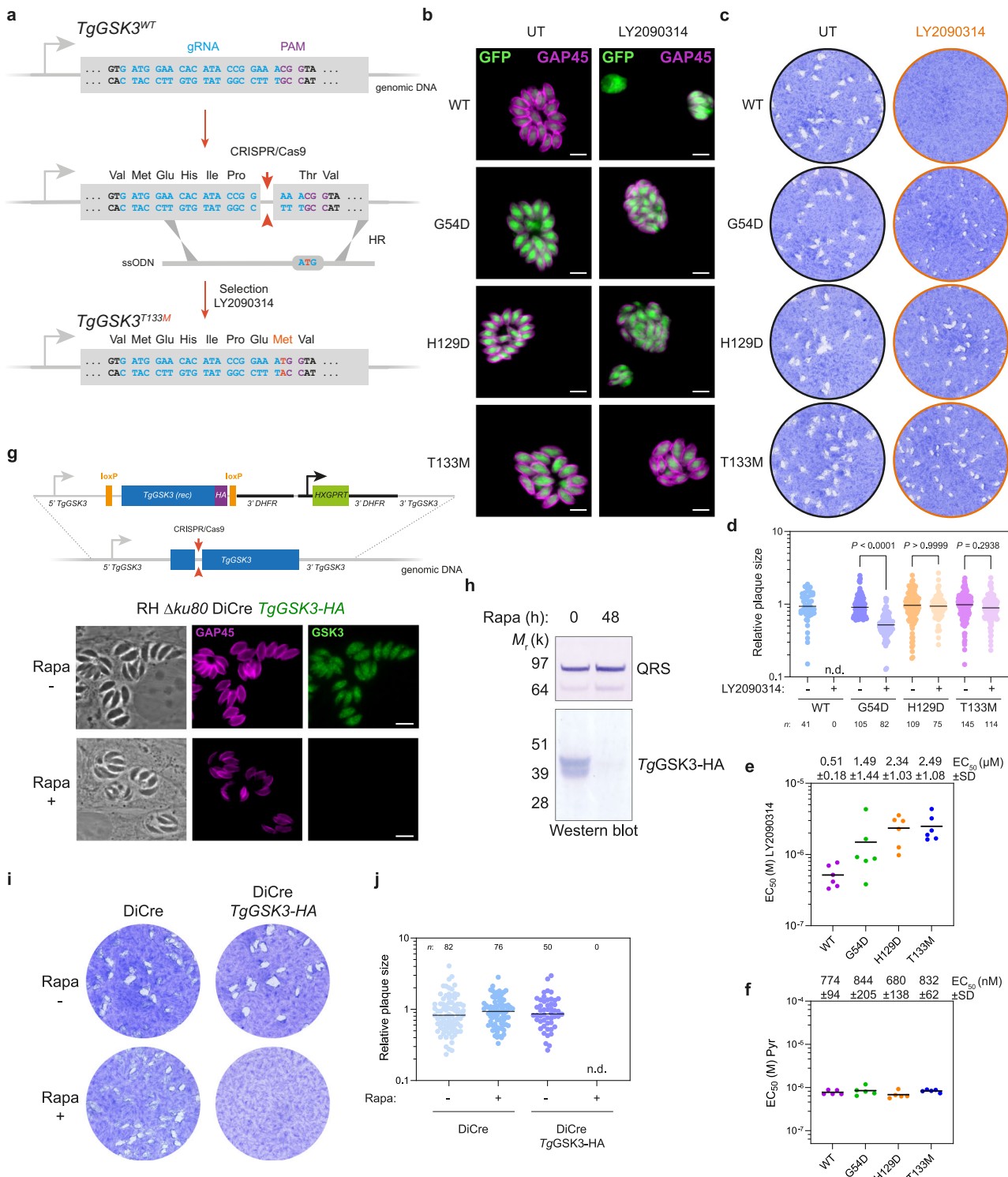

## Discussion

In this study, we report the identification of LY2090314 as a potent growth inhibitor of *Toxoplasma gondii*, showing remarkable potency at sub-micromolar concentrations, low cytotoxicity to host cells and promising broad-spectrum potential against apicomplexan pathogens, including *Cryptosporidium parvum*. Specifically, LY2090314 demonstrated activity against *C. parvum* with an EC$_{50}$ ranging from 160 to 286 nM, representing a more than 2000-fold improvement compared to paromomycin, a drug known for its anti-*Cryptosporidium* activity[30].

Using a forward genetic approach combining transcriptome sequencing with computational mutation discovery, we identified

mutations in the *Tg*GSK3 coding sequence of *T. gondii* that confer resistance to LY2090314. Structural studies, including the determination of the first X-ray crystal structure of LY2090314 bound to *Tg*GSK3, complemented by functional validation using an in vitro kinase assay with purified components, confirmed that LY2090314 directly inhibits *Tg*GSK3 enzymatic activity. Collectively, these findings represent gold-standard evidence in the field of target deconvolution for bioactive small molecules and provide unequivocal confirmation for target engagement.

Importantly, we identified a T133M substitution in *Tg*GSK3 that confers resistance to LY2090314. This threonine residue is highly

**Fig. 3 | Validation of *Tg*GSK3 as the Target of LY2090314. a** Schematic of the *TgGSK3* gene editing strategy in *T. gondii* parasites. Detailed view of *TgGSK3* locus and CRISPR/Cas9-mediated homology-directed repair with single-stranded oligo DNA nucleotides (ssODNs) carrying nucleotide substitutions (orange letters). After homologous recombination (HR) events with ssODNs, *TgGSK3* recombinants were selected with LY2090314. The engineered parasites were then validated by Sanger sequencing (see Supplementary Fig. 3a). **b** Fluorescence microscopy showing intracellular growth of wild-type (WT, RH *ku80* NLuc) and the *TgGSK3*-edited parasites (RH *ku80* NLuc *Tg*GSK3$^{G54D}$, RH *ku80* NLuc *Tg*GSK3$^{H129D}$, and RH *ku80* NLuc *Tg*GSK3$^{T133M}$). HFF cells were infected with tachyzoites of the indicated *T. gondii* strains expressing the NLuc-P2A-EmGFP reporter gene and incubated with 600 nM LY2090314 or 0.1% DMSO as control. Cells were fixed 24 h post-infection and then stained with antibodies against the *T. gondii* inner membrane complex protein GAP45 (magenta). The cytosolic EmGFP is shown in green. Scale bars, 5 μm. **c** Effects of *Tg*GSK3 mutations on *T. gondii* lytic cycle as assessed by plaque assay. Plaque sizes were measured after 7 days of growth in the presence or absence of 600 nM LY2090314. **d** Quantification of plaque sizes of wild-type (WT) and *TgGSK3*-edited parasites cultured with and without LY2090314. n.d., not detected. *p* values corresponding to Kruskal-Wallis test with Dunn's multiple comparisons with the untreated (without LY2090314) conditions are indicated. Data are mean from *n* = 3

independent experiments. **e, f** EC$_{50}$ values for LY2090314 (**e**) and pyrimethamine (Pyr) (**f**) were determined for WT and the *TgGSK3*-edited strains (G54D, H129D, and T133M). Data represent the mean from *n* = 6 and *n* = 5 independent biological replicates, each performed with three technical replicates, respectively. Mean EC$_{50}$ values ± SD are indicated. **g** Schematic representation of the DiCre/loxP-based inducible knock-out strategy used for the conditional depletion of *Tg*GSK3. The endogenous *TgGSK3* locus of recombinant parasites (RH DiCre *TgGSK3*) is flanked by loxP sites (orange) and tagged with HA (purple) at the C-terminus. rec, recodonized. Below, fluorescence microscopy of the RH DiCre *TgGSK3* strain after 48 h of treatment with DMSO or 50 nM rapamycin (Rapa). Cells were fixed 24 h post-infection and then stained with antibodies against GAP45 (magenta) and anti-HA antibodies (green). Scale bar, 5 μm. **h** Validation of Rapa-dependent depletion of *Tg*GSK3-HA in the RH DiCre *TgGSK3* strain by Western blotting analysis with anti-HA antibody. Cells were treated as in (**g**). *Tg*QRS was used as loading control. **i** Plaque assays of the indicated strains after 7 days of growth. Cells were left untreated or treated with 50 nM Rapa. **j** Quantification of plaque area of *n* = 3 independent infections, calculated from at least 50 plaques and normalized to the mean plaque area of the parental strain. n.d., not detected. Source data are provided as a Source Data file.

conserved across homologous enzymes, including the human GSK3, as well as orthologs from *Cryptosporidium* and multiple other Apicomplexan species. The high prevalence of the T133M mutation in our forward genetic screens, coupled with the significant resistance it conferred, suggests that this mutation could represent the primary mechanism of resistance in patients treated with LY2090314 or related chemotypes. These findings not only emphasize the need to develop next-generation inhibitors to circumvent resistance, but also provide a blueprint for the development of structurally robust compounds targeting resistant variants of *Tg*GSK3.

Using a forward genetic approach, we identified several mutations conferring drug resistance, all of which localize within or near the small-molecule binding site of *Tg*GSK3. Notably, no mutations were detected elsewhere that could modulate kinase activity through allosteric mechanisms. Converging evidence from yeast two-hybrid assays and crystallographic analyses revealed covalent dimerization of *Tg*GSK3, mediated by a conserved cysteine residue within the activation loop. This intermolecular interaction potentially represents a broader regulatory mechanism conserved within a subset of protein kinases. Moreover, this dimerization mechanism, which parallels the redox-regulated dimerization of human Aurora kinase, suggests that *Tg*GSK3 activation can be similarly modulated by redox signaling. While this hypothesis is conceptually appealing, the physiological relevance of this dimeric state in *T. gondii* remains to be established. Further biochemical validation, including targeted mutagenesis (e.g., Cys-to-Ser substitutions) and enzymatic assays on purified monomeric and dimeric species, will be necessary to clarify whether this covalent dimerization contributes to *Tg*GSK3 activation and/or regulation in vivo.

*Tg*GSK3 emerges as a critical kinase in *T. gondii*, validated through a combination of chemical genetics, conditional mutagenesis, and structural biology. While the *Plasmodium falciparum* ortholog *Pf*GSK3 has been explored as a potential antimalarial target[31–35], *Tg*GSK3 represents the first GSK3 ortholog in Apicomplexa for which both the drug-target relationship and atomic-resolution structure have been elucidated. Notably, the chemical scaffolds of the *Pf*GSK3 inhibitors reported in these studies are structurally distinct from LY2090314, suggesting that LY2090314 represents a novel pharmacological class with anti-apicomplexan activity. Functional inactivation of *Pf*GSK3β (PF3D7_0312400) leads to a moderate reduction in erythrocyte invasion and impaired gametocyte maturation, without affecting overall parasite viability during asexual blood stages. This contrasts with our findings in *T. gondii*, where *Tg*GSK3 inhibition by LY2090314 or its conditional depletion results in profound growth arrest and abnormal

daughter cell formation in both tachyzoite and bradyzoite stages, highlighting potential species-specific functions for GSK3 in Apicomplexa. The *Tg*GSK3 protein is also predicted to be expressed in non-tachyzoite stages based on transcriptomic data (ToxoDB), though its essentiality during the enteroepithelial sexual stages in the definitive host remains to be determined. These functional comparisons suggest that while Apicomplexan GSK3 enzymes may share conserved roles in parasite development and host cell invasion, *Tg*GSK3 likely fulfills additional lineage-specific functions, possibly through its interaction with apical complex components and cytoskeletal regulators, as revealed by our proximity labeling and yeast two-hybrid analyses.

From a structural perspective, comparative analysis of *Tg*GSK3 with its human ortholog *Hs*GSK3β reveals key differences at the ATP-binding site, particularly within the gatekeeper and hinge regions (Supplementary Fig. 2b). In *Tg*GSK3, residues M127 and H129 correspond to leucine and tyrosine in *Hs*GSK3β, respectively, and these substitutions lie within the inhibitor coordination interface as revealed by our 2.1 Å crystal structure with LY2090314. Similar strategies for achieving selectivity were previously explored in *Plasmodium* GSK3 by docking and homology modeling[31], underscoring the potential of leveraging subtle structural divergences to design selective GSK3 inhibitors against Apicomplexan pathogens.

Collectively, our findings establish *Tg*GSK3 as the molecular target of LY2090314 and elucidate its mechanism of inhibition, providing a strong rationale for the continued exploration of Apicomplexan GSK3 as a therapeutic target.

## Methods

### Parasite strains and human cell cultures

Human primary fibroblasts (HFFs, ATCC® CCL-171™) were cultured in Dulbecco's Modified Eagle Medium (DMEM) (Invitrogen) supplemented with 10% heat-inactivated fetal bovine serum (FBS) (Invitrogen), 10 mM (4-(2-hydroxyethyl)-1-piperazine-ethanesulfonic acid) (HEPES) buffer pH 7.2, 2 mM L-glutamine and 50 μg/ml penicillin and streptomycin (Invitrogen). Human ileocecal adenocarcinoma cells (HCT-8) cultured in RPMI 1640 with glutamine supplemented with 10% fetal bovine serum, 1 mM sodium pyruvate, penicillin (50 U/mL), and streptomycin (50 μg/mL). Cells were incubated at 37 °C with 5% CO$_2$ in humidified air. *Toxoplasma* strains used in this study and listed in Supplementary Data 1 were maintained in vitro by serial passage on monolayers of HFFs. Cultures were free of mycoplasma as determined by qualitative PCR.

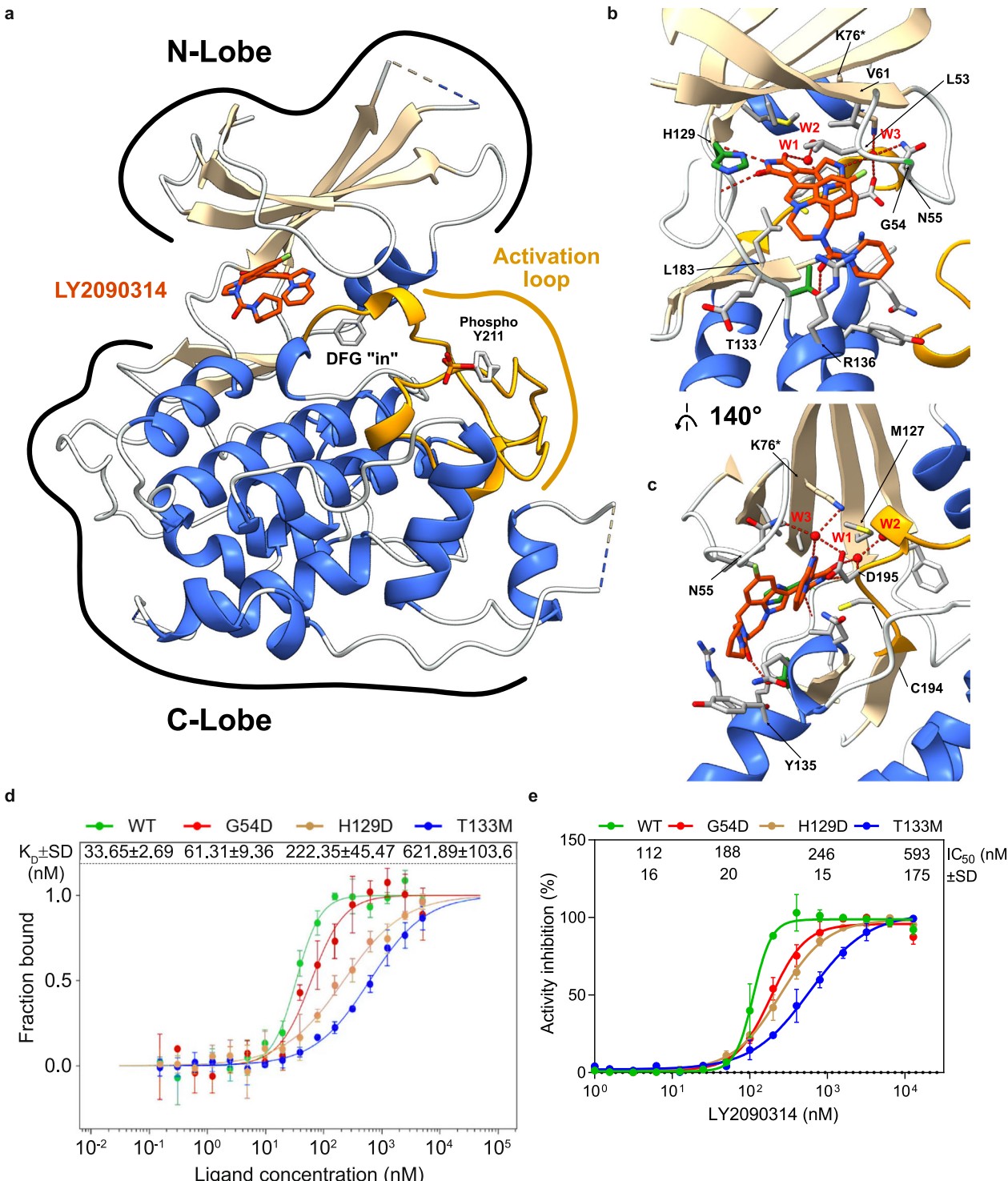

**Fig. 4 | Structural basis of LY2090314 binding and inhibition of *Tg*GSK3.**
**a** Overall crystallographic structure of *Tg*GSK3 bound to LY2090314. The cartoon representation shows alpha helices in dodger blue and beta-strand in tan while the activation loop is separately colored in gold. The LY2090314 inhibitor is colored in orange and displayed in a stick representation. F196 (from the DFG motif) and phosphorylated Y211 sidechains are also displayed in a grey stick representation. N- and C-lobes are indicated. **b** Focus on the LY2090314 binding site. Using the same color scheme as in A, side chains involved in the compound interactions are displayed, with resistance conferring residues are shown in green while hydrogen bonds are displayed in dotted red lines and water molecules are shown as red spheres. **c** Rotation of the B representation by 140°. **d** LY2090314 binding affinity titration determined using microscale thermophoresis on wild-type and mutant *Tg*GSK3. Normalized binding ratios are plotted against the tested LY2090314 concentrations and fitted with a Hill-Langmuir equation (used for cooperative binding models) to determine the apparent $K_d$ ($K_D$). WT, G54D, H129D and T133M values are plotted, respectively, in green, red, brown and blue. Data are presented as means ± SD from $n = 3$ independent experiments. **e** Endpoint kinase activity inhibition by LY2090314 on wild-type and mutant recombinant *Tg*GSK3. WT, G54D, H129D and T133M values are plotted respectively in green, red, brown and blue. Data are presented as means ± SD from $n = 3$ independent experiments. Dose response curves were fitted to calculate the $IC_{50}$ values. Source data are provided as a Source Data file.

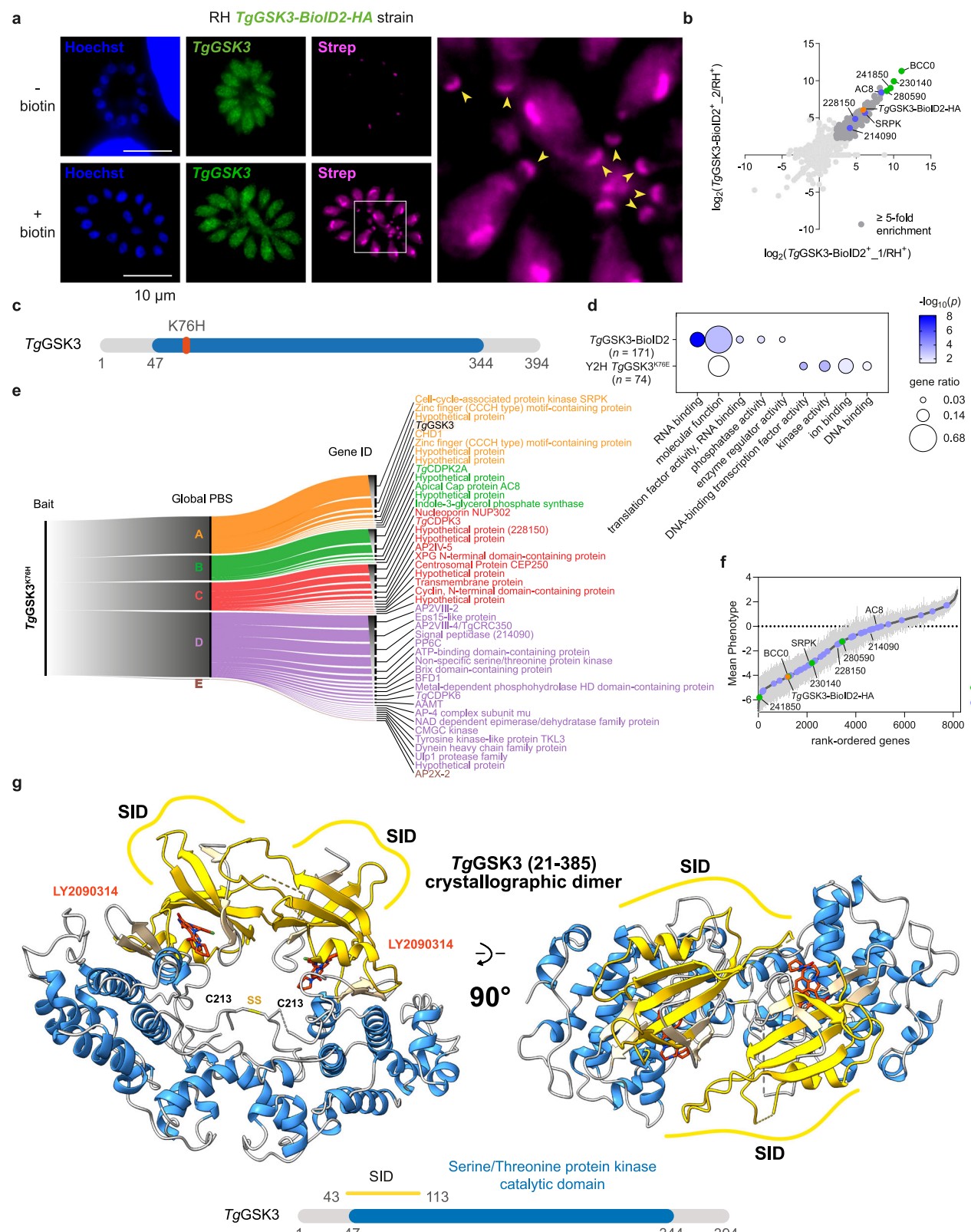

## Reagents

The compounds LY2090314 and Indirubin were purchased from TargetMol and Sigma-Aldrich, respectively.

The following primary antibodies were used in the immunofluorescence, and/or immunoblotting: rabbit anti-*Tg*GAP45 (gift from Pr. Dominique Soldati), mouse anti-HA tag (Roche, RRID: AB_2314622), rabbit anti-HA Tag (Cell Signaling Technology, RRID: AB_1549585),

rabbit anti-FLAG (Cell Signaling Technology, RRID: AB_2798687) and mouse anti-MYC clone 9B11 (RRID: AB_2148465). They were manufactured by Eurogentec and used for immunofluorescence and immunoblotting. Secondary immunofluorescent antibodies were coupled with Alexa Fluor 488 or Alexa Fluor 594 (Thermo Fisher Scientific). Secondary antibodies used in Western blotting were conjugated to alkaline phosphatase (Promega).

**Fig. 5 | Identification of *Tg*GSK3-interacting proteins. a** Immunofluorescence microscopy showing *Tg*GSK3 endogenously tagged with BioID2-HA (*Tg*GSK3-BioID2-HA) in intracellularly growing tachyzoites. Biotinylated proteins were detected using fluorophore-conjugated streptavidin (Strep). HFF cells were infected with tachyzoites (RH *Tg*GSK3-BioID2-HA) and incubated in the presence or absence of 150 µM biotin for 24 h prior to cell fixation and stained with antibodies against the HA tag (green), streptavidin-AlexaFluor 594 (magenta), and Hoechst (blue) to detect nuclei. Scale bars, 10 µm. **b** Scatter plot of the log2(ratio) of the proteins found in the two replicates (samples 1 and 2). The *Tg*GSK3-BioID2-HA fusion protein is shown in orange, the four most highly enriched proteins are labelled in green, the candidates also identified by the yeast two-hybrid approach are indicated in blue, the proteins with at least a 5-fold enrichment in the two biological replicates are in dark grey, and the other identified proteins in light grey. **c** Schematic representation of the *Tg*GSK3[K76H] kinase-dead mutant used as a bait in the yeast two-hybrid screening of the *T. gondii* cDNA library. The conserved GSK3 catalytic domains (blue) and the position of the K76E mutated residue (orange bar) are shown. **d** Gene ontology (GO) enrichment analysis of the proteins identified by mass spectrometry in the proximity labeling assay (*Tg*GSK3-BioID2) and the yeast two-hybrid screening approach using *Tg*GSK3[K76E] as a bait. Gene ratio is the proportion of proteins with the indicated GO term divided by the total number of proteins. Significance was determined with a hypergeometric test providing a p-value for the probability that the observed enrichment occurred by chance; only GO terms with $p < 0.05$ are shown. Redundant GO terms were removed. **e** Alluvial diagram summarizing the Yeast-two hybrid screen using *Tg*GSK3[K76H] as a bait (left) and the identified partners (right) having the highest Global PBS scores. The flow diagram shows the aggregation of the hits identified into the five Global PBS score categories shown on the center. The dimension of rectangles on the right is proportional to the prevalence of phosphorylated residues for each protein (ToxoDB). **f** *T. gondii* genes rank-ordered according to their fitness scores, based on data from [23] and ToxoDB.org. Genes identified as interactors by yeast two-hybrid (Y2H, blue) or BioID2 (green) approaches, as shown in panel (B), are highlighted. The *Tg*GSK3-BioID2-HA fusion protein is indicated in orange. Data are presented as mean ± SEM from $n = 4$ independent experiments. (**g**) Overall crystallographic structure of disulfide linked dimer of *Tg*GSK3 obtained from the monoclinic crystal at 3.0 Å. The cartoon representation shows alpha helices in blue and beta-strands of the N-lobes in gold while the disulfide linked between the two C213 residues in the activation loops are also displayed in a grey and gold stick representation (SS). The LY2090314 inhibitor is colored in green and displayed in a stick representation. The Selected Interaction Domain (SID) region determined by the Y2H approach are indicated. Below, schematic representation of *Tg*GSK3 protein architecture. Multiple independent interacting prey clones ($n = 17$, cell-cycle-associated protein kinase GSK, putative, Supplementary Data 4) allowed SID (in yellow) analysis that delineates the shortest fragment that is shared with all the interacting clones, and thus represents a potential region mediating the *Tg*GSK3-*Tg*GSK3 interaction. Source data are provided as a Source Data file.

## Drug screening analysis

The primary screening was conducted using a curated library of 514 small molecules, each provided as a 1 mM stock solution in dimethyl sulfoxide (DMSO)[18]. Compounds were screened in triplicate at a final concentration of 5 µM with 0.5% DMSO. *T. gondii* RH strain expressing NanoLuc luciferase (NLuc) and Emerald GFP (EmGFP) (RH NLuc) was used to enable quantitative assessment of parasite burden via fluorescence and luminescence readouts. Percentage growth inhibition was calculated relative to the DMSO vehicle control (set to 100% growth) as described previously[18]. For each compound, a composite z-score was calculated using the formula: z-score = (% inhibition − mean) / standard deviation. DMSO-treated wells and pyrimethamine (positive control) were included on each plate as internal references. Parasite load was quantified by EmGFP fluorescence, while host cell viability was assessed in parallel by Hoechst staining. This dual-readout system enabled simultaneous evaluation of antiparasitic activity and cytotoxicity (Supplementary Data 1).

## Measurement of EC$_{50}$ for *T. gondii* parasites and CC$_{50}$ determination

The in vitro inhibitory activity of small compounds on *Toxoplasma gondii* proliferation was assessed using the following protocol. A total of 2000 tachyzoites of the *T. gondii* RH strain expressing nanoluciferase (RH NLuc) were allowed to invade confluent human foreskin fibroblast (HFF) monolayers in a 96-well plate for 2 h. The inhibitor LY2090314, along with pyrimethamine (as a minimum signal control), were diluted in growth medium and added to the monolayers at various concentrations in triplicate (technical replicates), with DMSO-treated controls serving as maximum signal controls. The assay was conducted in a final volume of 100 µL.

After 48 h of incubation at 37 °C, the medium was removed and 50 µL of PBS was added to each well. The NanoLuc assays were performed using the Nano-Glo® Luciferase Assay System according to the manufacturer's instructions (Promega). Lysis was achieved by adding 50 µL of Nano-Glo® Luciferase Assay Reagent containing a 1:50 dilution of Nano-Glo® Luciferase Assay Substrate. Following a 3-minute incubation, luminescence was measured using the CLARIOstar® plate reader (BMG Labtech). Bioluminescence values from uninfected host cells were used to determine the background signal.

EC$_{50}$ values were calculated using non-linear regression analysis of normalized data, assuming a sigmoidal dose-response curve. The EC$_{50}$ values for each compound represent the average of at least three independent biological replicates. The cytotoxicity of LY2090314 on HFF and ARPE-19 cells was evaluated after 72 h of incubation using the CellTox™ Green Cytotoxicity Assay and CellTiter-Blue® Reagent (Promega).

## Measurement of EC$_{50}$ for *C. parvum* parasites and CC$_{50}$ determination for HCT-8

HCT-8 cells were grown to 80% confluency in white 96-well plates (Thermo Scientific Nunc MicroWell) and infected with freshly purified nLuc-mCherry-−expressing oocysts (multiplicity of infection (MOI), 0.5). LY2090314 and Indirubin stock solutions were prepared at 100 mM in DMSO. After 3 h, infected cell cultures were washed 3 times, and medium was replaced with LY2090314 or Indirubin derivate E804 at different concentrations and further incubated for 48 h. ¾ of the culture supernatant was removed from the wells (six replicates for each concentration), and 25 µL of Nano-Glo lysis buffer containing 1:50 of Nano-Glow substrate (Promega) was added to the wells. After 3 min of incubation, luminescence was measured with GloMax-Multi+ (Promega) and EC$_{50}$ was calculated using GraphPad Prism software from the dose-response inhibition curve. The cytotoxic concentration 50% (CC$_{50}$) of the compounds were tested at 48 h incubation using the MTS assay (CellTiter 96 Aqueous One Solution Cell Proliferation Assay, MTS, Promega).

## Plasmids and primers

Oligonucleotides were ordered from Sigma-Aldrich. PCR amplifications were performed with KOD Xtreme™ Hot Start DNA Polymerase. Primers and plasmids used or generated in this study are listed in Supplementary Data 1.

The bicistronic vectors expressing the Cas9 genome editing enzyme and specific sgRNAs targeting the *Tg*GSK3 coding sequence were constructed as described previously[36]. Briefly, oligonucleotides TgGSK3[G54D]-CRISPR-FWD and TgGSK3[G54D]-CRISPR-REV, TgGSK3[H129D]-CRISPR-FWD and TgGSK3[H129D]-CRISPR-REV, and TgGSK3[T133M]-CRISPR-FWD and TgGSK3[T133M]-CRISPR-REV (Supplementary Data 1) were annealed and ligated into the pTOXO_Cas9-CRISPR plasmid to create vectors used for construction of *T. gondii* recombinant for TgGSK3[G54D], TgGSK3[H129D] and TgGSK3[T133M], respectively.

To construct the vectors pLIC-TgGSK3-HF-DHFR, pLIC-TgGSK3-mAID-HA-DHFR, and pLIC-TgGSK3-BioID2-HA-HXGPRT the coding sequence of *Tg*GSK3 was amplified using primers LICF-265330_F and LICR-265330_R with RH *ku80* genomic DNA as the template. The PCR product was then cloned into the pLIC-HF-DHFR, pLIC-mAID-HA-

DHFR, and pLIC-BioID2-HA-HXGPRT (GenBank: PQ679940) vectors, respectively, using the LIC cloning method as described by ref. 37.

To generate the *TgGSK3* conditional knock out (cKO) plasmid, pUC57Simple-LoxP-GSK3-LoxP-HXGPRT, the *TgGSK3* 5′UTR with a loxP site inserted 100 bp upstream of the *TgGSK3* start codon, and a recodonized *TgGSK3* cDNA-HA sequence, were DNA-synthesized by GenScript (GenBank: PQ181256) using plasmid pUC57Simple-LoxP-HXGPRT (Genbank: PQ154619) as vector backbone.

Vectors for recombinant expression of *TgGSK3* (aa 21-385 and aa 46-385) in insect cell were codon optimized and synthetized by Genscript (GenBank: PQ846733 and PQ846732, respectively) using the pFastBac1 vector as backbone.

### *Toxoplasma* transfection and generation of parasite lines
*T. gondii* strains were transfected using electroporation parameters established previously[18]. Stable integrants or recombinants were selected with 25 mg/ml mycophenolic acid and 50 mg/ml xanthine or 1 mM pyrimethamine, and cloned by limiting dilution.

For endogenous gene tagging in *Toxoplasma*, the RH *ku80* strain was used as described[37]. Briefly, ~20 μg of each construct was linearized within the region of homology. Linearized vectors pLIC-TgGSK3-HF and pLIC-TgGSK3-BioID2-HA-HXGPRT (EcoRV) were phenol-chloroform extracted and ethanol precipitated. Constructions were re-suspended in TlowE buffer (10 mM Tris-HCl [pH8.0], 0.1 mM EDTA) for transfection. Stable recombinants were selected with pyrimethamine or mycophenolic acid and xanthine, single-cloned by limiting dilution, and verified by IFA. The resulting strains are RH *TgGSK3-HF* and RH *TgGSK3-BioID2-HA*, respectively (Supplementary Data 1).

To generate the conditional *TgGSK3* knock-out strain (RH DiCre *TgGSK3*), the plasmid pUC57Simple-LoxP-GSK3-LoxP-HXGPRT was linearized using NdeI and cotransfected with pTOXO_Cas9-CRISPR::sgTgGSK3$^{H129D}$ into the RH DiCre strain[24]. Recombinant parasites were selected 24 h post transfection by addition of 25 mg/mL mycophenolic acid and 50 mg/mL xanthine to culture medium. Resistant parasites were cloned, and successful integration at the *TgGSK3* locus and absence of the endogenous sequenced were confirmed using primer pairs P1/P2 and P1/P3. Rapamycin-induced excision of the floxed *TgGSK3* sequence was confirmed using primer pair P1/P4.

### *C. parvum* transfection and generation of parasite lines
*C. parvum* INRAE and IOWA (Univ. of Arizona) strains were transfected using an AMAXA Nucleofector 4D device (Lonza) with plasmids generated by Wilke et al[38]. and purchased from Addgene (pACT1:Cas9-GFP, U6:sgTK (Cat# 122852); Plasmid: pACT1:Cas9-GFP, U6:sgUPRT (Cat# 122853). For every transfection, $1.2 \times 10^8$ purified *C. parvum* WT oocysts were suspended in 1 mL of 0,75% taurocholic acid suspension in PBS and incubated at 37 °C for ~45 minutes. The progress of excystation was checked under microscope and excysted sporozoites were washed in PBS. The parasites were then suspended in 95 μL of SE buffer (SF cell Line 4D-Nucleofector kit, Lonza) and mixed with 100 μg plasmids in a final volume of 100 μL before being electroporated in a nucleocuvette vessel (Lonza). Following the electroporation, 100 μL of PBS was immediately added and 100 μL of the suspension inoculated orally to each of the two GKO mice that had received 5 min earlier 100 μL of an 8% sodium bicarbonate buffer. The mice were submitted to paromomycin selection at 16 mg/mL in the drinking water from 24 h post infection. The two *C. parvum* strains (INRAE-nLuc-mCherry and IOWA-nLuc-mCherry) were verified for mCherry expression by IFA and Nluc activity (Supplementary Data 1).

### Immunofluorescence microscopy
Cells cultured on coverslips were fixed in 3% formaldehyde for 20 minutes at room temperature, permeabilized with 0.1% (v/v) Triton X-100 for 15 minutes, and blocked in phosphate-buffered saline (PBS)

containing 3% (w/v) bovine serum albumin (BSA). Samples were incubated for 1 h with primary antibodies (GAP45), followed by secondary antibodies conjugated to Alexa Fluor 350, 488 or 594 (Molecular Probes) to detect intracellular parasites. Nuclei were stained with Hoechst 33258 for 10 minutes at room temperature. Coverslips were mounted on glass slides using Mowiol mounting medium, and 0.25 μm Z-stack images were acquired with an Axio Imager M2 fluorescence microscope (Carl Zeiss, Inc.). Images were processed using Icy 2.0 software (icy.bioimageanalysis.org) with the EpiDEMIC plugin for blind deconvolution of each channel separately. Maximum projections of deconvoluted stack images are presented.

### Western blot
Immunoblot analysis of protein was performed as described in ref. 18. Briefly, ~10$^7$ cells were lysed in 50 μL lysis buffer (10 mM Tris-HCl, pH6.8, 0.5% SDS [v/v], 10% glycerol [v/v], 1 mM EDTA and protease inhibitors cocktail) and sonicated. The protein extracts were separated by SDS-PAGE, and transferred to a PolyVinyliDene Fluoride membrane (PVDF; immobilon-P, Millipore) by liquid transfer. Membranes were then blocked with PBS buffer containing 0.01% Tween 20 (v/v) and 5% nonfat dry milk. Appropriate primary antibodies diluted in PBS containing 0.03% Tween 20 (v/v) were used to probe the membrane. Primary antibodies were detected using alkaline phosphatase conjugated secondary antibodies and the nitro blue tetrazolium (NBT)/5-Bromo-4-chloro-3-indolyl phosphate (BCIP) revelation (Amresco) or enhanced chemiluminescence system (Thermo Fisher Scientific).

### Plaque assays
Parasite growth was assessed following the protocol described by ref. 22. Briefly, freshly egressed parasites were inoculated onto a confluent monolayer of HFFs and cultured for 7 days with or without the indicated compounds. Cells were then fixed and stained with Coomassie blue staining solution (0.1% Coomassie R-250 in 40% ethanol and 10% acetic acid). Plaque sizes, when present, were measured using ZEN 2 lite software (Carl Zeiss Inc.) or Icy 2.0 software (icy.bioimageanalysis.org) with the Magic Wand Tool to select plaques.

### *Toxoplasma gondii* random mutagenesis
Parasites were chemically mutagenized following the protocol described by ref. 22, with the following modifications. ~10$^7$ tachyzoites (RH strain) growing intracellularly in HFF cells within a T25 flask were incubated at 37 °C for 4 h in 0.1% Fetal Bovine Serum (FBS) DMEM growth medium containing either 2.5 mM ethyl methane sulphonate (EMS) or the appropriate vehicle controls. Post-mutagen exposure, parasites were washed three times with PBS. The mutagenized population was then allowed to recover in a fresh T25 flask containing an HFF monolayer without the drug for 3–5 days. Subsequently, released tachyzoites were inoculated into fresh cell monolayers in a medium containing 600 nM LY2090314 and incubated until viable extracellular tachyzoites emerged 8–10 days later. Surviving parasites were passaged once more under continued LY2090314 treatment and cloned by limiting dilution. Three cloned mutants were isolated from each of the 8 independent mutagenesis experiments, resulting in unique SNV pools for each flask.

### RNA-seq, sequence alignment, and variant calling
For each biological assay, a T175 flask containing a confluent monolayer of HFF was infected with RH wild-type or LY2090314-resistant lines. Total RNA was extracted and purified using TRIzol (Invitrogen, Carlsbad, CA, USA) and RNeasy Plus Mini Kit (Qiagen). RNA quantity and quality were measured using NanoDrop 2000 (Thermo Scientific). RNA sequencing was performed as previously described by Bellini et al[22]., following standard Illumina protocols, by Novogene (Munich, Germany). Briefly, RNA quantity, integrity, and purity were determined using the Agilent 5400 Fragment Analyzer System (Agilent

Technologies, Palo Alto, California, USA). The integrity values or RIN ranged from 9.1 to 10 for all samples, which was considered sufficient. Messenger RNAs (mRNA) were purified from total RNA using poly-T oligo-attached magnetic beads. After fragmentation, the first strand cDNA was synthesized using random hexamer primers. For directional libraries, the second strand cDNA was synthesized using dUTP, instead of dTTP. The directional library was ready after end repair, A-tailing, adapter ligation, size selection, USER enzyme digestion, amplification, and purification. The library was checked with Qubit and real-time PCR for quantification and bioanalyzer for size distribution detection. Quantified libraries were pooled and sequenced on Illumina platforms, according to effective library concentration and data amount. The samples were sequenced using the Illumina NovaSeq platform, employing both non-stranded (for variant calling analysis) and strand-specific sequencing (for gene expression analysis) with $2 \times 150$ bp reads and generated ~40 million paired-end reads for each sample.

For variant calling analysis (non-stranded libraries), the RNA-Seq reads (FASTQ files) were processed and analyzed using the Lasergene Genomics Suite version 15 (DNASTAR, Madison, WI, USA) using default parameters. The paired-end reads were uploaded onto the SeqMan NGen (version 17, DNASTAR. Madison, WI, USA) platform for reference-based assembly and variant calling using the Toxoplasma Type I GT1 strain (ToxoDB-52, GT1 genome) as a reference template. The ArrayStar module (version 17, DNASTAR. Madison, WI, USA) was used for normalization, variant detection, and statistical analysis of uniquely mapped paired-end reads using the default parameters. Variant calls were filtered to select variants in coding regions with the following criteria: variant depth $\geq 30$, Qcall $\geq 60$, and absence in the parental wild-type strain (Supplementary Data 2). SNVs, insertions, and deletions in regulatory or intergenic regions were filtered out as they are unlikely to contribute to drug resistance. Mutations were plotted on a Circos plot using Circa (OMGenomics.com). The Illumina RNA-seq dataset generated during this study is available at National Center for Biotechnology Information Gene Expression Omnibus (NCBI GEO): GSE286338.

## Bradyzoite differentiation and drug treatment assays

Differentiation of tachyzoites into bradyzoites was induced by $CO_2$ depletion at physiological pH (7.4) using HFF as host cells. HFF monolayers were maintained in $CO_2$ Independent Medium (Gibco, Cat# 18045-088) supplemented with 5% FBS, 2mM L-glutamine, and 50 µg/mL penicillin−streptomycin (Invitrogen). For stage conversion, we employed the ME49 pGRA1-dsRed2.0 pBAG1-mNeonGreen reporter strain[39], which constitutively expresses the red fluorescent protein (RFP) dsRed2.0 under the control of the tachyzoite-specific GRA1 promoter and expresses mNeonGreen (mNG) from the bradyzoite-specific BAG1 promoter. Infected cultures were incubated at 37°C under ambient $CO_2$ conditions to promote spontaneous differentiation. At 24 h post-infection (hpi), infected monolayers were washed with PBS to remove extracellular parasites, and the medium was replaced. Cultures were subsequently maintained by refreshing the medium every 48 h and washing with PBS once per week. After 14 days of differentiation in $CO_2$ Independent Medium under ambient $CO_2$, monolayers were trypsinized, and the resulting cell suspensions were resuspended in Hank's Balanced Salt Solution (HBSS) supplemented with 5% FBS. mNeonGreen-positive cysts were isolated by fluorescence-activated cell sorting (FACS) using a CytoFLEX SRT Cell Sorter (Beckman Coulter), and 200 sorted cysts were seeded per well in a 24-well plate containing fresh $CO_2$-Independent Medium (Supplementary Fig. 1f-g).

Twenty-four h post-sorting, cultures were treated with either DMSO (vehicle control), 2 µM pyrimethamine, or 600 nM LY2090314. Following 72 h of incubation under bradyzoite-inducing conditions, cells were fixed in 4% paraformaldehyde and analyzed by immunofluorescence microscopy.

## T. gondii genome editing

Targeted genome modifications were performed using the CRISPR/ Cas9 system as described previously[18]. The recombinant parasites harboring allelic replacement for $TgGSK3^{G54D}$, $TgGSK3^{H129D}$, and $TgGSK3^{T133M}$ were generated by electroporation of the *T. gondii* RH NLuc strain with pTOXO_Cas9-CRISPR vectors targeting the $Tg$GSK3 coding sequence (pTOXO_Cas9-CRISPR::sgTgGSK3$^{G54D}$, pTOXO_Cas9-CRISPR::sgTgGSK3$^{H129D}$, and pTOXO_Cas9-CRISPR::sgTgGSK3$^{T133M}$) and their respective donor single-stranded oligo DNA nucleotides (ssODNs) carrying respective nucleotide substitutions (TgGSK3G54D_donor, TgGSK3H129D_donor, and TgGSK3T133M_donor; Supplementary Data 1) for homology directed repair. Recombinant parasites were selected with 600 nM LY2090314, as described previously in ref. 18 prior to subcloning by limited dilution, and allelic replacement was verified by sequencing of *T. gondii* GSK3 genomic DNA.

## Targeted nanopore sequencing

Parasites were chemically mutagenized as described above. ~3×10^7 tachyzoites (RH strain) growing intracellularly in HFF cells were incubated at 37 °C for 4 h in 0.1% Fetal Bovine Serum (FBS) DMEM growth medium containing either 2.5 mM ethyl methane sulphonate (EMS) or the appropriate vehicle controls. Post-mutagen exposure, parasites were washed three times with PBS. The mutagenized population was then allowed to recover in a fresh T25 flask containing an HFF monolayer without the drug for 3−5 days. Subsequently, released tachyzoites were inoculated into fresh cell monolayers in a medium containing 600 nM LY2090314 and incubated until viable extracellular tachyzoites emerged 8−10 days later. The resistant parasites were harvested and stored at -80 °C.

DNA was extracted from tachyzoites using the Blood & Cell Culture DNA Midi kit (Qiagen, 13343) and the Genomic-tip 100/G (Qiagen, 102343). Purified and unsheared high-molecular weight genomic DNA was resuspended in TE buffer (10 mM Tris-HCl, 1 mM EDTA, pH 8.0) and stored at 4 °C until use. DNA was quantified using the Qubit fluorometer (Thermo) immediately before performing the assay.

Guide RNAs were assembled as a duplex from synthetic Alt-R® CRISPR-Cas9 crRNA, (IDT, custom designed) and Alt-R® CRISPR-Cas9 tracrRNA (IDT, 1072532). Sequences are provided in Supplementary Data 1. The crRNAs were designed using CHOPCHOP (chopchop.c-bu.uib.no) design tool and selected for the highest predicted on-target performance with minimal significant secondary structure and off-target activity. The gRNA duplex was designed to introduce cuts on complementary strands flanking the region of interest. The gRNAs were designed to flank the TGGT1_265330 coding region; the target size between gRNAs was 7410−7773 kb.

Prior to guide RNA assembly, all crRNAs were pooled into an equimolar mix, with a total concentration of 100 µM. The crRNA mix and tracrRNA were then combined in nuclease free duplex buffer (IDT, 11-01-03-01) such that the tracrRNA concentration and total crRNA concentration were both 10 µM. The gRNA duplexes were formed by denaturation for 5 minutes at 95 °C, then allowed to cool to room temp for 5 minutes on a benchtop. Ribonucleoprotein complexes (RNPs) were constructed by combining 100 pmol of gRNA duplexes with 49.6 pmol of HiFi Cas9 Nuclease V3 (IDT, 1081060) in reaction buffer (RB) from the Oxford Nanopore Technologies (ONT) Cas9 Sequencing Kit (SQK-CS9109) at a final volume of 100 µL, incubated 30 minutes at room temperature, then stored at 4 °C until use, up to a week.

Five µg of high-molecular weight genomic DNA was dephosphorylated in 30 µL RB and 3 µL of Phosphatase (ONT, SQK-CS9109) for 10 min at 37 °C, followed by heating for 2 minutes at 80 °C for enzyme inactivation. After allowing the sample to return to room temperature, 10 µL of the pre-assembled Cas9/gRNA complexes was added to the sample. In the same tube, 1 µL of 10 mM dATP and 1 µL of Taq DNA polymerase (ONT, SQK-CS9109) were added for cleaving of

target DNA and dA-tailing of DNA ends. The sample was then incubated at 37 °C for 15 minutes for Cas9 cleavage followed by 5 minutes at 72 °C for dA-tailing. Sequencing adaptors (AMX, ONT Cas9 Sequencing Kit) were ligated to DNA ends using T4 Ligase and the ligation buffer (LNB). Prior to purification, SPRI Dilution Buffer (SDB) was added to the sample. The purification step involved the addition of 0.3X AMPure XP beads (Beckman Coulter, A63881), followed by two washes on a magnetic rack using long fragment buffer (LFB) to ensure selective enrichment of DNA fragments ≥3 kilobase pairs. DNA library was eluted in 25 μL of elution buffer (EB, ONT Cas9 Sequencing Kit). Sequencing libraries were prepared by adding the following to the eluate: 75 μL sequencing buffer (SQB,) and 51 μL loading beads (LB, ONT Cas9 Sequencing Kit).

Samples were run on a MinION (ver 9.4.1) flow cell using the MK1B. Sequencing runs were operated using the MinKNOW software (v24.06.16).

Basecalling was performed using GUPPY (Version 6.4.6) to generate FASTQ sequencing reads. Reads were aligned to the *T. gondii* reference genome (ToxoDB-67, GT1 genome) using Minimap2 (v2.28). Alignment files were converted to bam files and sorted using samtools.

### Recombinant expression of *Tg*GSK3 using baculovirus

Bacmid cloning steps and baculovirus generation were performed using EMBacY baculovirus (gifted by I. Berger), which contains a yellow fluorescent protein reporter gene in the virus backbone. The established standard cloning and transfection protocols set up within the EMBL Grenoble eukaryotic expression facility were used. Although baculovirus synthesis (V0) and amplification (to V1) were performed with SF21 cells cultured in SF900 III medium (Life Technologies), large-scale expression cultures were performed with Hi-5 cells cultured in Express-Five medium (Life Technologies) and infected with 0.5% (v/v) of generation 2 (V1) baculovirus suspensions and harvested 72 h after infection.

### *Tg*GSK3 expression and purification

Eight cell pellets, each from a 250 mL Hi5 culture, were resuspended in 120 mL of lysis buffer [50 mM tris (pH 8.0), 300 mM NaCl and 2 mM β-mercaptoethanol (β-ME)] in the presence of an anti-protease cocktail (Complete EDTA-free, Roche) and 4 μl of benzonase (Merck Millipore, 70746). Lysis was carried out on ice using sonication for 4 minutes (30-s on/ 30-s off, 45% amplitude). Clarification was then achieved by centrifugation at 12,000 g for 1 h at 4 °C. Following this, 5% glycerol and 20 mM imidazole were added and the mixture was incubated with 6 ml of Ni–nitrilotriacetic acid (NTA) resin (QIAGEN) at 4 °C for 30 min using a stirring magnet. All subsequent purification steps were conducted at room temperature. Once pulled into the column, the resin was washed twice with 10 cv of lysis buffer containing 20 mM imidazole and 5% glycerol with an intermediate wash using 1 M NaCl. Elution was performed by increasing lysis buffer imidazole concentration up to 300 mM. Eluted fractions were pooled based on SDS-PAGE analysis and passed through a pre-equilibrated [in 100 mM NaCl, 50 mM tris (pH 7.5), 1 mM β-ME, and 5% glycerol] heparin column connected to an AKTA Pure system. *Tg*GSK3 does not bind to the column, but many contaminents are retained. Last, the recovered heparin flow through was concentrated and injected onto a SUPERDEX 200 increase 10/300GL (GE Healthcare) column operating in 300 mM NaCl, 50 mM Tris pH8, 1 mM β-ME and 1% glycerol, with elution monitored at 280 nm. Peak fractions were concentrated using a 30-kDa Amicon concentrator, then frozen in liquid nitrogen and stored long-term at −80 °C.

### Size exclusion chromatography coupled to laser light scattering (SEC-MALLS)

*Tg*GSK3 (amino acids 21-385) SEC-MALLS was performed on a S200 10/ 300 GL increase column (GE Healthcare) running in a buffer system containing 20 mM tris (pH 7.5), 300 mM NaCl, and 1 mM β-ME on an OMNISEC (Malvern) system equipped with RALS, LALS, UV RI and Viscosimeter detectors. Injections of 50 μl were performed using protein samples concentrated at a minimum of 2 mg/mL, and a constant flow rate of 0.5 mL/min was used. Data treatment and mass determination were performed using the OMNISEC software.

### Microscale thermophoresis

Microscale thermophoresis (MST) measurements were performed using a NanoTemper Monolith NT.115 Green/Red instrument. 200 nM WT and mutated *Tg*GSK3 proteins were added to 100 nM dye solution from the Monolith His-Tag Labeling Kit RED-tris-NTA second generation (NanoTemper Technologies) both diluted in 50 mM NaCl, 20 mM Tris HCl pH8, 1 mM MgCl$_2$ and 0.05% Tween 200. After a 30-minute incubation at room temperature, the samples were centrifuged to clear residual compound. In parallel, a series of 14 twofold dilutions of LY2090314 starting from 5 μM were prepared in pure DMSO. Finally, the labeled protein and drug dilution mixture (v/v) were loaded into premium capillaries (NanoTemper Technologies) for measurement using the instrument's recommended settings of 40% light-emitting diode power and 40% MST power.

### Kinase activity assays

Serial dilution of LY2090314 were prepared from 500 μM to 30 nM in pure DMSO. One microliter of this serial dilution and pure DMSO for control was added to 39 μL of 150 nM *Tg*GSK3, 25 μM ATP and 10 μM GSM peptide substrate (RRRPASVPPSPSLSRHS(pS)HQRR; synthetized by EMD Millipore Corporation), within 50 mM NaCl, 50 mM tris (pH 7.5), 1 mM β-ME, 1% BSA and 5 mM MgCl$_2$. The reaction mix was kept for 3 h at 30 °C in a thermocycler. The final ATP content was determined using the Kinase-Glo Plus assay (Promega) according to the manufacturer's protocol and luminescence titrations were measured on the CLARIOstar (BMG Labtech) with a 10-second acquisition time within white 96-well plates. Raw luminescence data were normalized for each condition, with the lowest value (control with 100% DMSO) set to represent 100% kinase activity. All titrations were performed in triplicate, with the entire experiment repeated three times for consistency. Apparent IC$_{50}$ determination was realized through a four-parameter nonlinear inhibition curve fitting in GraphPad Prism.

### Crystallization with LY2090314

5 mg/ml of *Tg*GSK3 (aa 46-385) was incubated with 1 mM LY2090314 for 5 minutes at room temperature and centrifuge to eliminate compound aggregate prior injection onto an S200 columns running into 300 mM NaCl, 50 mM Tris pH8, 1 mM β-ME and 1% glycerol. The eluted protein fractions were pooled and concentrated to a final concentration of 20 mg/ml. Crystallization was performed using the hanging drop vapor diffusion method with GSK3/LY2090314 (1:1) with 1 M sodium malonate pH 6. Crystals appearing generally after few days were harvested using Hampton nylon loops, cryoprotected in 6 M sodium malonate and flash-frozen in liquid nitrogen. With a similar approach, the second construct yielding the monoclinic crystal system (aa 20 to 385) was also prepared with LY2090314 and hanging drops were setup in 5% MPD, 9% PEG 6000 and 0.1 M Tris pH 7.5. Crystals only started growing after several weeks with these conditions. Cryoprotection of these crystals was performed with precipitant supplemented by 20 % glycerol.

### Data collection and structure determination

X-ray diffraction data for *Tg*GSK3/LY2090314 crystals were collected by the autonomous European Synchrotron Radiation Facility (ESRF) beamline MASSIF-1[40,41] using automatic protocols for the location and optimal centering of crystals[42]. In all cases, diffraction was performed at 100 K. Data reduction was per- formed using XDS[43] while amplitude

scaling/merging was performed by Xscale. The first molecular replacement solution for the orthorhombic crystal was obtained with Phaser[44] using the crystal AlphaFold model of *Tg*GSK3 [AlphaFold/Uniprot A0A125YY75 entry] as a template. The initial phasing solution was then improved through cycles of manual adjustment in Coot[45], automated building in phenix autobuild[46] and refined using Refmac5 or phenix refine. Final pdb model corrections were performed using the pdb-redo server[47]. A similar approach was also used for the monoclinic crystal though the initial orthorhombic crystal structure of *Tg*GSK3 was used as molecular replacement search model.

## Yeast two-hybrid analysis

Yeast two-hybrid screening was performed by Hybrigenics Services, S.A.S., Evry, France (http://www.hybrigenics-services.com). The coding sequence for *Tg*GSK3$^{K76H}$ (aa 1-394) was PCR-amplified from plasmid pUC57Simple-*Tg*GSK3$^{K76H}$ (GenBank: PQ181257) and cloned into pB29 vector as an N-terminal fusion to LexA (*Tg*GSK3$^{K76H}$-LexA). The construct was checked by sequencing the entire insert and used as a bait to screen a random-primed *Toxoplasma gondii* wild-type RH strain cDNA library constructed into pP6 vector. pB29 and pP6 derive from the original pBTM116[48,49] and pGADGH[50] plasmids, respectively. Parasite RNA for the *Toxoplasma gondii* Y2H library was generated by Drs. Sherri Huang and Bill Sullivan at the Indiana University School of Medicine. 47.4 million clones ( ~ 5-fold the complexity of the library) were screened using a mating approach with YHGX13 (Y187 ade2-101::loxP-kanMX-loxP, matα) and L40ΔGal4 (MATa) yeast strains as previously described[51]. 246 His$^+$ colonies were selected on a medium lacking tryptophan, leucine and histidine, and supplemented with 10 mM 3-aminotriazole to handle bait autoactivation. The prey fragments of the positive clones were amplified by PCR and sequenced at their 5' and 3' junctions. The resulting sequences were used to identify the corresponding interacting proteins in the GenBank database (NCBI) using a fully automated procedure. A confidence score (PBS, for Predicted Biological Score) was attributed to each interaction as previously described[52].

The PBS relies on two different levels of analysis. Firstly, a local score takes into account the redundancy and independency of prey fragments, as well as the distribution of reading frames and stop codons in overlapping fragments. Secondly, a global score takes into account the interactions found in all the screens performed at Hybrigenics using the same library. This global score represents the probability of an interaction being nonspecific. For practical use, the scores were divided into four categories, from A (highest confidence) to D (lowest confidence). A fifth category (E) specifically flags interactions involving highly connected prey domains previously found several times in screens performed on libraries derived from the same organism. Finally, several of these highly connected domains have been confirmed as false-positives of the technique and are now tagged as F. The PBS scores have been shown to positively correlate with the biological significance of interactions[53,54].

## Proximity labeling of intracellular *Tg*GSK3-BioID2-HA strain and mass spectrometry-based proteomics

Freshly egressed parasites expressing *Tg*GSK3-BioID2-HA (RH *TgGSK3-BioID2-HA* strain) were used to infect confluent HFF monolayers in two 175 cm² flasks per condition ( ~ 10⁷ parasites per flask). The parental RH *ku80* strain was used as a control. The parasites invaded and replicated in host cells for 24 h in the presence or absence of 150 μM biotin (Sigma-Aldrich, B4639). Parasites invaded and replicated in host cells for 24 h in the presence or absence of 150 μM biotin (Sigma-Aldrich, B4639). Following incubation, parasites were harvested, washed three times with 50 mL of cold PBS, and stored at -80 °C until further processing. Experimental duplicates were performed for *Tg*GSK3-BioID2-HA samples.

Parasites were lysed in 400 μL radioimmunoprecipitation assay (RIPA) buffer (50 mM Tris [pH 8.0], 150 mM NaCl, 0.1% SDS, 0.5% sodium deoxycholate, 1% Triton X-100) supplemented with complete protease inhibitor cocktail (Roche) and incubated on ice for 30 minutes. Samples were sonicated (Diagenode Bioruptor, 5 cycles of 15 s ON/30 s OFF), yielding a total protein content of 2.1 ± 0.3 mg. Insoluble debris was removed by centrifugation (14,000 × g, 30 minutes), and the supernatants were incubated with 200 μL of magnetic streptavidin beads (Bio-Adembeads, 300 nm) pre-equilibrated twice in RIPA buffer. Incubation was performed for 2 h at room temperature with gentle agitation (5 rpm).

Following incubation, the beads were washed sequentially: twice with 1 mL RIPA buffer containing 1 mM EDTA (cold; 10 minutes), once with 1 mL 1 M KCl (at room temperature; 10 minutes), once with 1 mL 0.1 M Na₂CO₃ (cold; 1 minute), once with 1 mL 2 M urea in 10 mM Tris-HCl (pH 8.0; at room temperature, 1 minute), and twice with 1 mL RIPA buffer (cold; 10 minutes). The supernatant was discarded, and the beads were snap-frozen in liquid nitrogen and stored at -80 °C. Proteins were digested on beads using modified trypsin (Promega, sequencing grade), as described in ref. 55. The resulting peptides were analyzed by online nanoliquid chromatography coupled to MS/MS (Ultimate 3000 RSLCnano and Q-Exactive HF, Thermo Fisher Scientific). For this purpose, the peptides were sampled on a precolumn (300 μm × 5 mm PepMap C18, Thermo Scientific) and separated in a 75 μm × 250 mm C18 column (Aurora Generation 3, 1.7 μm, IonOpticks) using a 120 min acetonitrile gradient. The MS and MS/MS data were acquired by Xcalibur version 2.9 (Thermo Fisher Scientific). Peptides and proteins were identified by Mascot (version 2.8.3, Matrix Science) through concomitant searches against the *Toxoplasma gondii* database (ME49 taxonomy, version 58 downloaded from ToxoDB), the Uniprot database (*Homo sapiens* taxonomy, 202402 version), the sequence of *Tg*GSK3-BioID2-HA, and a homemade database containing the sequences of classical contaminant proteins found in proteomic analyses (human keratins, trypsin…). Trypsin/P was chosen as the enzyme and two missed cleavages were allowed. Precursor and fragment mass error tolerances were set at respectively at 10 and 20 ppm. Peptide modifications allowed during the search were: Carbamidomethyl (C, fixed), Acetyl (Protein N-term, variable) and Oxidation (M, variable). The Proline software ([56], version 2.3.1) was used for the compilation, grouping, and filtering of the results (conservation of rank 1 peptides, peptide length ≥6 amino acids, false discovery rate of peptide-spectrum-match identifications <1%[57], and minimum of one specific peptide per identified protein group). Proline was then used to perform a MS1 label-free quantification of the identified protein groups based on razor and specific peptides. Protein abundances were then compared between parasites expressing *Tg*GSK3-BioID2-HA and parasites that do not express any BioID2 fusion protein that were incubated with 150 μM biotin for 24 h. For this, only proteins identified in the *Toxoplasma gondii* database were considered, abundances were log transformed and median normalized, and missing values were imputed by the mean of the first percentile values of each sample. Proteins were considered proximal to *Tg*GSK3 if they were enriched at least 5 times in both *Tg*GSK3-BioID2-HA replicates compared to mock control.

## Enrichment analysis

Sets of gene ontology (GO) terms associated with proteins identified through proximity labeling (*Tg*GSK3-BioID2-HA) and yeast two-hybrid screening using *Tg*GSK3$^{K76E}$ as bait, along with the background proteome (all proteins quantified in each experiment), were retrieved from ToxoDB.org. GO terms were analyzed under the category "Molecular Function" with computed evidence, applying a *p* value cutoff of 0.05. Enrichment analysis was performed by comparing the GO terms identified in the proximity labeling and yeast two-hybrid datasets against those found in the background proteome. Statistical significance for enrichment was determined using a hypergeometric test, yielding *p* values that represent the likelihood of observed enrichments occurring by chance.

## Analysis of alphafold-multimer predictions

AlphaFold-Multimer structure predictions were analyzed using a custom Python script based on the AlphaPulldown framework[58], to extract interface features. For each predicted protein complex (provided as a.zip archive), the script parsed the model structure (.cif) and confidence metrics (.json), including the inter-protein predicted TM-score (iPTM). Chain–chain interfaces were identified based on a 5 Å distance cutoff between atoms. Only interactions with iPTM ≥0.6 were retained for further analysis. For each interface, residues from chain B (designated as the prey) involved in inter-chain contacts were extracted along with ±5 flanking residues. The resulting sequences were compiled and exported in tabular format for downstream analyses (Supplementary Data 5).

## Bioinformatic tools

Venn diagram[59].
 RAWGraphs Visualization Platform[60].
 Clustal Omega, iTOL[61].
 CLC Sequence Viewer (QIAGEN).
 DNAStar Lasergene v15 (DNASTAR, Madison, WI, USA).
 All models were depicted using UCSF ChimeraX[62].

## Reporting summary

Further information on research design is available in the Nature Portfolio Reporting Summary linked to this article.

## Data availability

All data needed to evaluate the conclusions in the paper are present in the paper and/or the Supplementary Materials. pUC57Simple-LoxP-HXGPRT: Genbank PQ154619 pUC57Simple-LoxP-GSK3-LoxP-HXGPRT: Genbank PQ181256 (PQ181256) pLIC-BioID2-HA-HXGPRT: GenBank PQ679940 (PQ679940) pUC57Simple-TgGSK3^K76H: GenBank PQ181257 (PQ181257) TgGSK3_aa21-385_cter6His_pFastBac1: Genbank PQ846733 (PQ846733) TgGSK3_aa46-385_cter6His_pFastBac1: Genbank PQ846732 (PQ846732) The Illumina RNA Sequencing data generated in this study have been deposited in the GEO Datasets under accession number GSE286338 The coordinates and structure factors for the TgGSK3/ LY2090314 structures have been deposited in the PDB with the accession number 9HVX and 9HW6. (https://www.rcsb.org/structure/9HVX and https://www.rcsb.org/structure/9HW6). Please note that residue numbering is offset by 21 amino acids due to the N-terminal truncation of the crystallized constructs. The mass spectrometry proteomics data have been deposited to the ProteomeXchange Consortium via the PRIDE[63] partner repository with the dataset identifier PXD059579 (https://www.ebi.ac.uk/pride/archive/projects/PXD059579). Source data are provided with this paper.

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

## Acknowledgements

We are grateful to the developers of the ToxoDB.org Genome Resource. ToxoDB and EuPathDB are part of the National Institutes of Health/National Institutes of Allergy and Infectious Diseases (NIH/NIAID)-funded Bioinformatics Resource Center. This work used the platforms of the Grenoble Instruct-ERIC center (ISBG; UAR 3518 CNRS-CEA-UGA-EMBL) within the Grenoble Partnership for Structural Biology (PSB), supported by FRISBI (ANR-10-INBS-0005-02) and GRAL, financed within the University Grenoble Alpes graduate school (Ecoles Universitaires de Recherche) CBH-EUR-GS (ANR-17-EURE-0003). We thank Aline Le Roy for assistance with the OMNISEC data collection. We acknowledge the European Synchrotron Radiation Facility for provision of beam time on ID30-A1/MASSIF-1. This work was supported by MSD Avenir [Project LatentToxoDiag, DS-2022-0017, M-A.H.], the Laboratoire d'Excellence

(LabEx) ParaFrap [ANR-11-LABX-0024, M-A.H.], the Agence Nationale pour la Recherche [Project ApiNewDrug, ANR-21-CE35-0010-01, M-A.H.; Project ApiMORCing [ANR-21-CE15-0002-01, M-A.H.]; Project ToxoP53 [ANR-19-CE15-0026, A.B.], Fondation pour la Recherche Médicale [FRM Equipe, EQU202103012571, M-A.H.], and the European Partnership Animal Health & Welfare-SOA19 [Grant Agreement No 101136346, F.L.]. MS-based proteomic experiments were partially supported by Agence Nationale de la Recherche under projects ProFI (Proteomics French Infrastructure, ANR-10-INBS-08, Y.C.) and GRAL, a program from the Chemistry Biology Health (CBH) Graduate School of University Grenoble Alpes [ANR-17-EURE-0003, Y.C.].

## Author contributions

Conceptualization: C.S., M-A.H., F.L., and A.B. Methodology: S.D-M., V.B., C.S., F.L., C.M., Y.C., V.B., M.B. and A.B. Investigation: S.D-M., C.S., V.B., I.D., J.W., M-P.B-P., A.T., L.B., C.C., C.M., M.B., and A.B. Funding acquisition: F.L., M-A.H. and A.B. Supervision: C.S., V.B., M-P.B-P., Y.C., F.L., M.B., M-A.H., and A.B. Writing—original draft: S.D-M., C.S., Y.C., F.L., M-A.H., and A.B. All authors discussed the results and commented on the manuscript.

## Competing interests

The authors declare that they have no competing interests.

## Additional information

[1]Institute for Advanced Biosciences (IAB), Team Host-pathogen interactions and immunity to infection, INSERM U1209, CNRS UMR5309, University Grenoble Alpes, Grenoble, France. [2]INRAE, Université François Rabelais de Tours, Centre Val de Loire, UMR1282 ISP, Laboratoire Apicomplexes et Immunité Mucosale, Nouzilly, France. [3]University Grenoble Alpes, CEA, INSERM, UA13 BGE, CNRS, CEA, Grenoble, France. [4]Integrated Structural Biology Grenoble (ISBG) CNRS, CEA, Université Grenoble Alpes, EMBL, 71 avenue des Martyrs, Grenoble, France. [5]European Molecular Biology Laboratory, Grenoble, 71 Avenue des Martyrs, CS 90181, Grenoble, France. [6]These authors contributed equally: Silvia Diaz-Martin, Christopher Swale. ✉e-mail: mohamed-ali.hakimi@inserm.fr; alexandre.bougdour@inserm.fr

