## [Transparent Peer Review file · Nature Communications]

Structural and Functional Characterization of *TgGSK3*, a Druggable Kinase in *Toxoplasma gondii*

Corresponding Author: Dr Alexandre BOUGDOUR

Version 0:

Reviewer comments:

Reviewer #1

(Remarks to the Author)

The manuscript by Diaz-Martin et al describes the identification and validation of the glycogen synthase kinase 3 (GSK3) in *Toxoplasma gondii* (Tg) as a druggable target. Notably, a human GSK3 inhibitor, LY2090314, discovered in a focused small-molecule screen of FDA-approved drugs, is leveraged as a chemical tool. A combination of induced drug-resistant parasites, biochemical characterization and structural biology are used to validate TgGSK3 as the drug target for LY2090314. This is most prominently highlighted in the determination of an atomic resolution co-crystal structure of LY2090314 bound to the ATP-binding site in recombinant TgGSK3. Biochemical studies are further used to determine binding affinities and validate resistance-conferring mutations acquired in parasite lines with reduced sensitivity to LY2090314. Finally, a two-step strategy was used to probe the interactome of TgGSK3: (1) a proximity labeling methodology, followed by (2) a yeast two-hybrid screen. The end result was suggestive of a conditional dimerization event that may be linked to the redox state of the cell.

Overall, the manuscript highlights a series of studies centered on identifying and characterizing TgGSK3 as a drug target. Those described studies are well executed and well described within the main text. However, a question of novelty, at least in the context of drug discovery, undermines the manuscript's premise that GSK3 is a novel target for Apicomplexa. Moreover, the manuscript lacks cohesion. It begins with a drug-centric exploration of TgGSK3, which is bolstered by nice X-ray crystal structure and enzymology, then the manuscript veers into the TgGSK3 interactome. However, it is not clear how the interactome studies relate back to TgGSK3 as a drug target. In fact, the interactome studies are absent from the Summary, which makes it feel insignificant/disconnected.

Major critiques:

1. The manuscript would benefit from either (1) refocus only on the target identification and validation studies and add further analysis of how the TgGSK3 enzyme diverges from human GSK3 (based on comparative crystal structures and residue conservation) and how that may present potential drug discovery opportunities or (2) retain the interactome studies and tie in how the identified interaction partners correlate with the drug-induced phenotype observed in the treated parasites. In *Plasmodium*, this gene is linked to invasion and maturation to sexual blood-stage parasites. How does this compare to Tg?
2. Given that GSK3 is known drug target for *Plasmodium* (not cited in the manuscript despite reports: PMID: 35961464 and 25861860) and *Cryptosporidium* (cited by the authors), then the current title should be revised from "Apicomplexan" to "*Toxoplasma gondii*".
3. It would be informative to include activity data for LY2090314 against the bradyzoite stage.
4. Consider consolidating panels C and F-J from Figure 1 into a separate table for easier viewing. The current plots could be moved to Supplementary section.

Additional suggestions:

1. Lines 103 and 104, please replace the terms "lower" with "more potent" to prevent any confusion from readers. The term "lower" can be interpreted two different ways.

2. Lines 154-156: It's unclear how the authors conclude a "greater fitness advantage" between mutants. That term is usually reserved to reflect differences in parasite replication and fitness which was not assessed here. The authors appear to use this term to reflect a higher degree of resistance for T133M and H129D over G54D. Please reword.
3. Line 172-173: This sentence reads awkwardly. Please consider changing to: "Overall, TgGSK3 has high sequence (54% identity) and structural conservation (r.m.s.d. = 1.67 Ang) to the human ortholog (Supplementary Figure 1B).
4. Line 216: Please add the term "steric" between "significant" and "clash".
5. Line 289: Do you mean "sub-micromolar" and not "sub-nanomolar"?
6. Throughout the Methods the subsections either end with a period or no period. Please standardize.
7. Please add additional details to the Drug Screening Analysis section in the Methods. Details, such as the final DMSO and drug concentrations, are not given. Also, it's a minor suggestion, but consider moving the corresponding assay description next to this section because both are linked.

Reviewer #2

(Remarks to the Author)

The manuscript reports identification of a GSK3 kinase analogue of Apicomplexan parasites as a novel target for treating diseases such as malaria, toxoplasmosis, and cryptosporidiosis. The identification of TgGSK3 as a critical enzyme for parasite's survival was done via chemical biology approach where phenotypic screen hits were followed up to identify their molecular targets. The target and a hit inhibitor were validated by a broad spectrum of assays from molecular to mice level. Overall, the manuscript is very clearly written, follows sound methodology and presents coherent evidence to support the claims. I consider it worthy publication in Nature Communications. However, I would like to suggest a few moderate improvements that might help to better explain and widen the scope of the manuscript:

1. The initial hit comes from a previous publication of the same authors (DOI : 10.1126/scitranslmed.abn3231) where another kinase inhibitor was studied. Neither current, nor the previous manuscript contains detailed description of the screen rationale. Particularly the choice of 514 molecules from approved medications (where I believe around 1500 are available) as well as particularly high hit ratio and the rationale for evaluation of these particular hits is not sufficiently rationalized. There is nothing wrong in re-using the results of previously performed screen to evaluate a new hit/target. However, this part of the manuscript seems to be overly brief and superficial.
2. Inhibitor LY2090314 from Eli-Lilly was discontinued due to safety issues, therefore it might not be very useful beyond chemical tool compound. However, there exist more advanced GSK3 inhibitors (e.g. Tideglusib) with improved safety profiles. Given the number of available molecules for both isoforms of GSK3 it would be very helpful to test such a molecule at least for in vitro TgGSK3 inhibition or perform structural modeling/analysis of binding. Hopefully, these, much safer molecules, can become applied antiparasitic drugs. There is a useful review of GSK inhibitors here: <https://doi.org/10.3389/fnmol.2021.792364>
3. In reference to the above, the authors show that the parasite can quite easily develop LY2090314 resistance by T133M mutation. This mutation occurs in the ATP binding cleft close to the hinge. The larger methionine side chain will clearly collide with the piperidine ring. A brief search in PDB I made show that for example YL-354 inhibitor might be insensitive to this mutation (<https://doi.org/10.1021/acscchemneuro.3c00135>). I suggest that some, close to the clinic, inhibitors of GSK3 should be evaluated for TgGSK3 binding and mutant resistance. This can be done by structural analysis or, ideally, binding or inhibition assay.
4. Interaction partners from Y2H ("SRPK, the Apical Cap protein AC8, two hypothetical proteins TGME49_228150 and TGME49_214090), offer an interesting opportunity to check for complex feasibility using AF3 predictions. Their interface compatibility can be easily checked and would prove great advance towards parasites' interactome.
5. The observed homodimerization propensity is a very interesting feature. The hypothesis on autophosphorylation derived from Aurora similarity seems plausible. In this light, I am not sure while the authors did not describe phosphorylation of Ser236 (numbering from CIF file) clearly seen in their monoclinic structure. This residue strongly contributes to the dimer formation by a salt bridge to Arg199. To clarify this, I suggest looking for the "in solution" dimerization state (size exclusion, mass photometry, NMR could be used). Both reducing and mild oxidative conditions can be used to evaluate these findings. I suggest more in-depth investigation here. Possibly also checking for dimers enzymatic activity and a Cys to Ser mutant.
6. The structures attached to the manuscript have different residue numbering that the one used in the manuscript. Thr133 is 114 in the file. This should be fixed for PDB deposition.

Reviewer #3

(Remarks to the Author)

This well-written manuscript describes the effect of LY2090314 (a small molecule originally developed as a human GSK3 inhibitor in the context of cancer chemotherapy) on *Toxoplasma gondii*, and proceeds with target deconvolution, identifying (perhaps unsurprisingly) the parasite's GSK3 orthologue as the target. This involved the generation of LY2090314-resistant parasites (the mutagenesis/selection/RNAseq approach is elegant), which identifies three mutations in TgGSK3 that confer resistance to the compound. That TgGSK3 not only mediates resistance, but is indeed the target of the compound, is

supported by the potent activity of the compound on the recombinant enzyme, and by structural data on the enzyme/ligand complex. An interactomics study is performed as well, which support the idea that TgGSK3 self-interacts (and provides some possible insight into the molecular roles of TgGSK3).

Overall, this paper proposes a well-supported novel compound-target pair in the context of urgently needed untapped targets to intervene against parasitic diseases. The data are compelling, with appropriate statistical validation. I have no major issue with the manuscript. The authors might want to consider the following minor points:

Line 71. "LY2090314, a maleimide-based kinase inhibitor, has advanced through phase 1/2 clinical trials for treating advanced solid tumors (15) and acute leukemia (16, 17)". Indicate here that LY2090314 has been developed as an inhibitor of human GSK3.

Line 128: "Notably, all identified mutations are confined to the serine/threonine protein kinase catalytic domain". Were there resistant clones carrying more than one of these mutations?

Line 129: Indicate where the T133 residue lies with respect to the kinase domain region (which subdomain? Is G45 the first G in the glycine triad that mediates (in part) ATP binding? Is the T133 one of the regulatory threonine? Are H129 and T333 close to the gatekeeper residue?

Line 151: "In genome-edited parasites, 151 the TGGT1_265330 mutations G54D, H129D, and T133M substantially reduced susceptibility to 152 LY2090314 compared to wild-type parasites". "Substantially" is vague and subjective. Please quantify this statement based on data in Fig 3.

Line 156: "Collectively, these data suggest that 157 LY2090314 directly interferes with TgGSK3 activity, resulting in impaired parasite growth". This is a bit of an over-statement – it cannot be excluded that the mutations in TgGSK3 may confer resistance through modulating its activity, even if the target is a different protein (although I agree that this is unlikely, in view of the fact that LY2090314 is a human GSK3 inhibitor, and of subsequent structural and enzyme inhibition data presented in the manuscript). Are there any data on resistance of human cells to the inhibitor? If so, it would be of great interest if that would map to the same (conserved) residues.

Line 290 "promising broad-spectrum potential against apicomplexan pathogens, including *Cryptosporidium parvum*". It would enhance the scope of this paper if the compound was shown to have activity against *Plasmodium falciparum* and *Plasmodium vivax* (in view of the need to develop antimalarials that are effective against both species, it would be useful to add *P. vivax* sequences to the phylogenetic tree in Fig. 2F).

Line 271: "cysteine-crosslinked kinase dimers are rare, with only a handful of crystal structures in the PDB available for comparison (including, but not limited to the PDB IDs: 1ZMW, 2WNT, 5NGO, 6VPI, 6PVJ)". It would be useful to name the kinases relating to these PDB IDs (only 6PVJ/Aurora kinase is named). Since the authors argue below that this may apply to entire CMGC group, it would be good to indicate that these are indeed members of the CMGC group.

Line 314: "This intermolecular interaction potentially represents a hallmark of CMGC kinase activation". Could this be this an overstatement? Is this Cys residue conserved within the entire CMGC group? Many kinases of this group have been crystallised -- has any such dimerization been observed with other members of the group? Aurora kinases (which are referred to as displaying a similar Cys-mediated dimerization) are not part of the CMGC group.

There are a number of papers describing the development of inhibitors against the *Plasmodium* orthologue PfGSK3 (see for example PMID: 36166733; PMID: 33813310; PMID: 23214499). It would be good to cite these papers, and mention if any of the compounds described therein have similarity with the series that includes LY2090314.

Version 1:

Reviewer comments:

Reviewer #1

(Remarks to the Author)

I commend the authors for their willingness to address and incorporate the collective reviewer feedback into the manuscript. Overall, the quality of the manuscript is excellent and I do not have any additional feedback.

Reviewer #3

(Remarks to the Author)

All comments and requests for clarification have been addressed satisfactorily.

REVIEWER COMMENTS

Reviewer #1 (Remarks to the Author):

The manuscript by Diaz-Martin et al describes the identification and validation of the glycogen synthase kinase 3 (GSK3) in *Toxoplasma gondii* (Tg) as a druggable target. Notably, a human GSK3 inhibitor, LY2090314, discovered in a focused small-molecule screen of FDA-approved drugs, is leveraged as a chemical tool. A combination of induced drug-resistant parasites, biochemical characterization and structural biology are used to validate TgGSK3 as the drug target for Ly2090314. This is most prominently highlighted in the determination of an atomic resolution co-crystal structure of LY2090314 bound to the ATP-binding site in recombinant TgGSK3. Biochemical studies are further used to determine binding affinities and validate resistance-conferring mutations acquired in parasite lines with reduced sensitivity to LY2090314. Finally, a two-step strategy was used to probe the interactome of TgGSK3: (1) a proximity labeling methodology, followed by (2) a yeast two-hybrid screen. The end result was suggestive of a conditional dimerization event that may be linked to the redox state of the cell.

Overall, the manuscript highlights a series of studies centered on identifying and characterizing TgGSK3 as a drug target. Those described studies are well executed and well described within the main text. However, a question of novelty, at least in the context of drug discovery, undermines the manuscripts premise that GSK3 is a novel target for Apicomplexa. Moreover, the manuscript lacks cohesion. It begins with a drug-centric exploration of TgGSK3, which is bolstered by nice X-ray crystal structure and enzymology, then the manuscript veers into the TgGSK3 interactome. However, it is not clear how the interactome studies relate back to TgGSK3 as a drug target. In fact, the interactome studies are absent from the Summary, which makes it feel insignificant/disconnected.

Major critiques:

1. The manuscript would benefit from either (1) refocus only on the target identification and validation studies and add further analysis of how the TgGSK3 enzyme diverges from human GSK3 (based on comparative crystal structures and residue conservation) and how that may present potential drug discovery opportunities or (2) retain the interactome studies and tie in how the identified interaction partners correlate with the drug-induced phenotype observed in the treated parasites. In *Plasmodium*, this gene is linked to invasion and maturation to sexual blood-stage parasites. How does this compare to Tg?

We agree with the reviewer that the link between the interactome and the drug-target validation needed strengthening. We have chosen to retain the interactome studies and now mentioned how the top interactors (e.g., SRPK, BCC0, TGME49_228150, and TGME49_241850) may contribute to TgGSK3-dependent phenotypes. To support this point, we have included a new panel (Figure 5F) presenting CRISPR-derived fitness scores for these candidate interactors. In addition, we have expanded the Results section (page 11, line 295-298) to address their potential roles in parasite proliferation. The Summary has also been revised to emphasize the relevance of the interactome studies and to address the concerns regarding the novelty of our findings.

Regarding the comparison between TgGSK3 and human GSK3 in terms of amino acid sequence and structural features, it remains difficult to speculate on potential avenues for drug selectivity or discovery,

as both enzymes are highly conserved at the sequence (Supplementary Figure 1B) and structural levels (Supplementary Figure 3B). Therefore, the selectivity observed *in cellulo* is more likely attributable to differences in the functional contribution of GSK3 activity to overall cellular fitness in the respective biological contexts (i.e., human fibroblasts versus *T. gondii* tachyzoites). This reflects the well-documented observation that GSK3 function and essentiality in human cells are context-dependent, varying according to cell type, developmental stage, and the presence of compensatory signaling mechanisms. To clarify this point and in light of prior findings in *Plasmodium*, we have added a dedicated paragraph to the Discussion section (page 13-14, lines 373-397), highlighting specific features of *Tg*GSK3 that diverge from its human counterpart. Additionally, Supplementary Figure 2B and its corresponding legend have been revised accordingly.

2. Given that GSK3 is known drug target for *Plasmodium* (not cited in the manuscript despite reports: PMID: 35961464 and 25861860) and *Cryptosporidium* (cited by the authors), then the current title should be revised from “Apicomplexan” to “*Toxoplasma gondii*”.

To reflect the organism-specific focus and to address concerns regarding novelty, the manuscript title has been revised to: “*Targeting TgGSK3 in Toxoplasma gondii: Structural and Functional Insights into a Conserved Apicomplexan Kinase with Drug Development Potential.*” As noted above, previous studies on GSK3 in *Plasmodium* have now been appropriately cited and discussed in the revised Discussion section (page 13-14, lines 373-397).

3. It would be informative to include activity data for LY2090314 against the bradyzoite stage. We thank the reviewer for this constructive suggestion, which addresses an important aspect of the compound’s activity. In response, we have assessed the activity of LY2090314 against the bradyzoite stage using an *in vitro* differentiation model based on CO₂-depletion. Infected HFF monolayers were cultured under bradyzoite-inducing conditions for 14 days using the ME49 pGRA1-dsRed2.0 pBAG1-mNeonGreen dual-reporter strain, which enables simultaneous monitoring of tachyzoite (GRA1 promoter-driven dsRed2.0) and bradyzoite (BAG1 promoter-driven mNeonGreen) gene expression. Following cell sorting of mNG-positive cysts, cultures were treated with 600 nM LY2090314 for 72 hours under bradyzoite-maintaining conditions.

Immunofluorescence analysis revealed a marked reduction in the expression of the bradyzoite markers BAG1, BCLA, and BSM, while expression from the GRA1 promoter was increased, suggesting a disruption of bradyzoite-specific gene expression. These effects were not observed following treatment with 2 μM pyrimethamine, which had only modest effects on either bradyzoite or tachyzoite markers. These findings are consistent with the documented expression of *Tg*GSK3 in bradyzoites (ToxoDB.org) and indicate that LY2090314 retains biological activity against this stage of the parasite. The corresponding results have been included in the revised manuscript and are shown in Supplementary Figure 1G–K. The Results section has been updated accordingly on page 5 (lines 115-126).

4. Consider consolidating panels C and F-J from Figure 1 into a separate table for easier viewing. The current plots could be moved to Supplementary section.

As suggested by the reviewer, we have consolidated the EC₅₀ and CC₅₀ values into summary tables now presented in the revised Figure 1C and 1G. The corresponding dose–response plots have been moved to a newly created Supplementary Figure 1 for improved readability and accessibility. Figure legend’s have been updated accordingly.

Additional suggestions:

1. Lines 103 and 104, please replace the terms “lower” with “more potent” to prevent any confusion from readers. The term “lower” can be interpreted two different ways.

As suggested by the reviewer, we have clarified the relationship between EC₅₀ values and compound potency in the revised manuscript (page 5, lines 128-130) by rephrasing the sentence to indicate that LY2090314 exhibits higher antiparasitic potency compared to *T. gondii* and indirubin E804, based on the observed EC₅₀ values.

2. Lines 154-156: It’s unclear how the authors conclude a “greater fitness advantage” between mutants. That term is usually reserved to reflect differences in parasite replication and fitness which was not assessed here. The authors appear to use this term to reflect a higher degree of resistance for T133M and H129D over G54D. Please reword.

We thank the reviewer for this clarification. Indeed, the term “fitness advantage” is not appropriate in this context, as we did not directly assess replication rates or competitive fitness between the mutants. In the revised manuscript (page 7, line 193), we have reworded the sentence to more accurately reflect the observed data. The text now reads: “This observation aligns with results from EMS-mutagenized parasites and further suggests that the T133M or H129D mutations confer a higher level of resistance to the compound compared to the G54D mutation (Figures 2B–C and 3B–E).”

3. Line 172-173: This sentence reads awkwardly. Please consider changing to: “Overall, TgGSK3 has high sequence (54% identity) and structural conservation (r.m.s.d. = 1.67 Ang) to the human ortholog (Supplementary Figure 1B).

We have corrected the sentence as suggested by the reviewer (page 8, lines 210-211).

4. Line 216: Please add the term “steric” between “significant” and “clash”.

Done.

5. Line 289: Do you mean “sub-micromolar” and not “sub-nanomolar”?

Corrected.

6. Throughout the Methods the subsections either end with a period or no period. Please standardize.

Corrected.

7. Please add additional details to the Drug Screening Analysis section in the Methods. Details, such as the final DMSO and drug concentrations, are not given. Also, it's a minor suggestion, but consider moving the corresponding assay description next to this section because both are linked.

The "Drug Screening Analysis" section in the *Materials and Methods* has been updated to provide additional details regarding compound concentrations used in the library screen as well as DMSO final concentration. In addition, and as suggested by the reviewer, the subsection describing the measurement of EC₅₀ values has been relocated to directly follow the "Drug Screening" section to improve logical flow and readability.

Reviewer #2 (Remarks to the Author):

The manuscript reports identification of a GSK3 kinase analogue of Apicomplexan parasites as a novel target for treating diseases such as malaria, toxoplasmosis, and cryptosporidiosis. The identification of TgGSK3 as a critical enzyme for parasite's survival was done via chemical biology approach where phenotypic screen hits were followed up to identify their molecular targets. The target and a hit inhibitor were validated by a broad spectrum of assays from molecular to mice level.

Overall, the manuscript is very clearly written, follows sound methodology and presents coherent evidence to support the claims. I consider it worthy publication in Nature Communications.

However, I would like to suggest a few moderate improvements that might help to better explain and widen the scope of the manuscript:

1. The initial hit comes from a previous publication of the same authors (DOI:10.1126/scitranslmed.abn3231) where another kinase inhibitor was studied. Neither current, nor the previous manuscript contains detailed description of the screen rationale. Particularly the choice of 514 molecules from approved medications (where I believe around 1500 are available) as well as particularly high hit ratio and the rationale for evaluation of these particular hits is not sufficiently rationalized. There is nothing wrong in re-using the results of previously performed screen to evaluate a new hit/target. However, this part of the manuscript seems to be overly brief and superficial.

We thank the reviewer for this thoughtful comment and welcome the opportunity to clarify the rationale behind our compound screening strategy and hit selection process.

The compound library used in our phenotypic screen comprises 514 pharmacologically annotated molecules, primarily consisting of FDA-approved drugs and late-stage clinical candidates. This library was originally obtained from TargetMol more than ten years ago and has since been maintained in our laboratory as a legacy screening set. At the time of acquisition, the subset was designed to cover a diverse range of therapeutic classes and druggable targets, with an emphasis on mechanistic diversity and compounds with known bioavailability and safety profiles. While larger and more comprehensive libraries of approved drugs (~1500 compounds) are now available, this particular collection is no longer commercially offered by the supplier. All hits emerging from this screen are systematically followed up, with priority given to compounds that show cross-activity against other apicomplexan parasites such as *Cryptosporidium parvum* or *Plasmodium falciparum*. The major bottleneck in advancing these hits lies in

target deconvolution. In several cases, resistant parasites cannot be recovered, or the targets turn out to involve well-characterized enzymatic pathways for which druggability has already been established. Such cases are generally not pursued further. The previously published study (DOI:10.1126/scitranslmed.abn3231) reported the characterization of a distinct compound from the same screen that targets *Tg*PRPK4. In the present study, we focus on LY2090314, a separate hit that we have functionally and structurally validated as targeting *Tg*GSK3, a kinase not previously characterized as a drug target in *T. gondii*. LY2090314 was prioritized for follow-up based on its nanomolar potency, strong selectivity index, cross-efficacy against *Cryptosporidium*, and the successful identification of resistance-conferring mutations that enabled biochemical target validation.

In response to the reviewer's suggestion, we have revised the Results section (page 4, lines 92–102) to provide additional details regarding the composition of the compound library, the rationale for its use in the phenotypic screen, and the criteria employed for hit prioritization. We also clarified how the present study builds upon data from our previously published screen by contextualizing the reuse of screening results and emphasizing that LY2090314 constitutes a distinct hit with a different target (*Tg*GSK3). We hope that this expanded explanation addresses the reviewer's concerns and strengthens the rationale and transparency of our screening strategy.

2. Inhibitor LY2090314 from Eli-Lilly was discontinued due to safety issues, therefore it might not be very useful beyond chemical tool compound. However, there exist more advanced GSK3 inhibitors (e.g. Tideglusib) with improved safety profiles. Given the number of available molecules for both isoforms of GSK3 it would be very helpful to test such a molecule at least for in vitro *Tg*GSK3 inhibition or perform structural modeling/analysis of binding. Hopefully, these, much safer molecules, can become applied antiparasitic drugs. There is a useful review of GSK inhibitors here: <https://doi.org/10.3389/fnmol.2021.792364>

We thank the reviewer for this insightful comment. In response, we have expanded the manuscript to clarify that several chemically distinct GSK3 inhibitors were present in the original compound library screened, including representatives of aminopyrimidine, imidazopyridine, and thiadiazolidinone classes. These compounds, however, did not show significant activity against *T. gondii* in our phenotypic assays, suggesting either insufficient engagement with the parasite kinase or limited intracellular access, possibly due to poor permeability across the parasitophorous vacuole membrane. The figure 1A has been update to highlight these compounds. To further explore the potential for *Tg*GSK3 inhibition with improved pharmacological candidates, we also tested three additional GSK3 inhibitors with favorable safety profiles (BRD3731, SAR502250, and Tideglusib), each at a concentration of 1 μ M. None of these compounds inhibited *T. gondii* or *Cryptosporidium* growth under the same assay conditions. These results, now described in the revised *Results* section (page 6, lines 134–142), support the conclusion that LY2090314 possesses a uniquely potent and selective inhibitory activity against apicomplexan parasites.

3. In reference to the above, the authors show that the parasite can quite easily develop LY2090314 resistance by T133M mutation. This mutation occurs in the ATP binding cleft close to the hinge. The larger methionine side chain will clearly collide with the piperidine ring. A brief search in PDB I made show that for example YL-354 inhibitor might be insensitive to this mutation (<https://doi.org/10.1021/acscchemneuro.3c00135>). I suggest that some, close to the clinic, inhibitors of

GSK3 should be evaluated for TgGSK3 binding and mutant resistance. This can be done by structural analysis or, ideally, binding or inhibition assay.

We thank the reviewer for examining the chemical incompatibility of the T133M mutant with LY2090314 and for proposing alternative strategies. However, before addressing this issue, we would like to clarify that although T133M represents the dominant resistance mutation, it is not readily acquired in cell culture and requires EMS-induced chemical mutagenesis. *Toxoplasma gondii* has a relatively low natural mutation rate (estimated at 10^{-9} to 10^{-10} per base per generation), and drug resistance is generally less of a concern in *T. gondii* compared to *Plasmodium falciparum*.

The reviewer is correct in suggesting that the YL-354 compound would likely circumvent the T133M mutation (as shown in the structural snapshots below). However, given that YL-354 differs substantially from LY2090314 and does not rely on a maleimide backbone, it is difficult to predict whether it would exhibit improved efficacy.

Our central point is that modifying the compound to bypass the T133M mutation is not straightforward, particularly if such structural changes compromise binding affinity or target specificity. Notably, the parent compound AT-7519, which is mentioned in the cited reference (<https://doi.org/10.1021/acchemneuro.3c00135>), corresponds to compound #158 in our screen and is primarily an inhibitor of cyclin-dependent kinases (CDKs) and CDK-like kinases. This molecule exhibited only modest inhibitory activity against *T. gondii* (35.69% inhibition of parasite proliferation) and was therefore not prioritized for further characterization. Nevertheless, we fully acknowledge the importance of investigating more advanced or structurally diverse GSK3 inhibitors. In response to this comment, as also noted above, we have now incorporated this perspective into the revised Results section (page 6, lines 134–142).

4. Interaction partners from Y2H (“SRPK, the Apical Cap protein AC8, two hypothetical proteins TGME49_228150 and TGME49_214090), offer an interesting opportunity to check for complex feasibility using AF3 predictions. Their interface compatibility can be easily checked and would prove great advance towards parasites’ interactome.

We thank the reviewer for this insightful suggestion. In response, we employed AlphaFold-Multimer (AF3) to assess the structural feasibility of interactions between *Tg*GSK3 and selected candidate proteins identified through both the BioID2 proximity labeling and yeast two-hybrid (Y2H) screens. Specifically, we evaluated SRPK, AC8, TGME49_228150, and TGME49_214090. While this integrative structural approach yielded high-confidence predictions for several interactors (now summarized in Supplementary Table 6), it did not support strong interface compatibility for the aforementioned proteins. This may reflect the inherent limitations of current structure prediction algorithms, particularly in modeling transient or low-affinity interactions such as those typically observed between kinases and their substrates.

We have revised the Results section accordingly (see page 10, lines 288–294) to include this information and to clarify that although AlphaFold-Multimer did not confirm stable interaction interfaces for these candidates, their identification by both BioID2 and Y2H still provides convergent experimental support for potential biological relevance. We agree that further investigation using orthogonal validation methods (e.g., co-immunoprecipitation, functional epistasis) will be essential to fully define these interactions. *Methods* were modified accordingly.

5. The observed homodimerization propensity is a very interesting feature. The hypothesis on autophosphorylation derived from Aurora similarity seems plausible. In this light, I am not sure while the authors did not describe phosphorylation of Ser236 (numbering from CIF file) clearly seen in their monoclinic structure. This residue strongly contributes to the dimer formation by a salt bridge to Arg199. To clarify this, I suggest looking for the “in solution” dimerization state (size exclusion, mass photometry, NMR could be used). Both reducing and mild oxidative conditions can be used to evaluate these findings. I suggest more in-depth investigation here. Possibly also checking for dimers enzymatic activity and a Cys to Ser mutant.

We thank the reviewer for this insightful and constructive comment regarding the disulfide-mediated homodimerization observed in the *Tg*GSK3 crystal structure. As suggested, we have further explored the oligomeric state of *Tg*GSK3 in solution under various redox conditions. SEC-MALS analysis performed under reducing conditions (5 mM β -mercaptoethanol) showed no evidence of *Tg*GSK3 dimerization in solution. We extended these investigations using both reducing (1 mM β -mercaptoethanol) and mildly oxidative conditions (1:1 ratio of reduced and oxidized glutathione at 1 mM each). Under all tested conditions, *Tg*GSK3 behaved as a monomer based on gel filtration and native PAGE analyses (see the figure below). These findings suggest that disulfide-mediated dimerization does not occur spontaneously in solution under the tested conditions and may require specific cellular factors or redox enzymes to be physiologically relevant.

We agree with the reviewer that the observation of a salt bridge between phosphorylated Ser236 and Arg199, clearly resolved in the monoclinic structure, may contribute to stabilizing the dimer interface as it likely rigidifies the terminal section of the activation loop (see chimerX snapshot below).

While we find this structural feature intriguing and potentially important in the context of autophosphorylation we have chosen not to highlight this feature as only 2 out of 4 GSK3 copies present this ptm, which makes this entire arrangement quite likely a crystallographic artifact. In contrast, the disulfide bond involving Cys213 is observed consistently across all subunits. We now mention this in the revised manuscript (Discussion section, page 13, lines 367–371) to temper our conclusions and highlight the need for additional studies, including mutagenesis (e.g., Cys213-to-Ser) and activity assays on defined dimeric states, to clarify the physiological relevance of this structural features which were observed.

6. The structures attached to the manuscript have different residue numbering than the one used in the manuscript. Thr133 is 114 in the file. This should be fixed for PDB deposition.

PDB reflects the recombinant protein numbering with a shift of 21 aa due to N-ter truncation, however your comment made us to recheck protein numbering within the figures and we noticed that a mistake (an increment of 10 aa) was made which concerns all the panels describing homodimeric description of *TgGSK3*. To harmonize, we always referred to a biological numbering. We have corrected C223 to C213 in Figure 5F and supplementary Figure 6. A note regarding the shift in numbering is now specified page 30, lines 858-859.

Reviewer #3 (Remarks to the Author):

This well-written manuscript describes the effect of LY2090314 (a small molecule originally developed as

a human GSK3 inhibitor in the context of cancer chemotherapy) on *Toxoplasma gondii*, and proceeds with target deconvolution, identifying (perhaps unsurprisingly) the parasite's GSK3 orthologue as the target. This involved the generation of LY2090314-resistant parasites (the mutagenesis/selection/RNaseq approach is elegant), which identifies three mutations in TgGSK3 that confer resistance to the compound. That TgGSK3 not only mediates resistance, but is indeed the target of the compound, is supported by the potent activity of the compound on the recombinant enzyme, and by structural data on the enzyme/ligand complex. An interactomics study is performed as well, which support the idea that TgGSK3 self-interacts (and provides some possible insight into the molecular roles of TgGSK3).

Overall, this paper proposes a well-supported novel compound-target pair in the context of urgently needed untapped targets to intervene against parasitic diseases. The data are compelling, with appropriate statistical validation. I have no major issue with the manuscript. The authors might want to consider the following minor points:

Line 71. "LY2090314, a maleimide-based kinase inhibitor, has advanced through phase 1/2 clinical trials for treating advanced solid tumors (15) and acute leukemia (16, 17)". Indicate here that LY2090314 has been developed as an inhibitor of human GSK3.

The text has been modified as suggested by the reviewer.

Line 128: "Notably, all identified mutations are confined to the serine/threonine protein kinase catalytic domain". Were there resistant clones carrying more than one of these mutations?

We thank the reviewer for this pertinent question. In the revised manuscript (page 6, lines 163–164), we have clarified that each resistant clone harbored only a single point mutation within the *TgGSK3* coding sequence.

Line 129: Indicate where the T133 residues lie with respect to the kinase domain region (which subdomain? Is G45 the first G in the glycine triad that mediates (in part) ATP binding? Is the T133 one of the regulatory threonine? Are H129 and T333 close to the gatekeeper residue?

The text has been modified as suggested by the reviewer (page 6, lines 161-162 of the revised version).

Line 151: "In genome-edited parasites, 151 the TGGT1_265330 mutations G54D, H129D, and T133M substantially reduced susceptibility to 152 LY2090314 compared to wild-type parasites". "Substantially" is vague and subjective. Please quantify this statement based on data in Fig 3.

We appreciate the reviewer's suggestion and agree that quantifying the degree of reduced susceptibility improves clarity and precision. In response, we have revised the corresponding sentence in the *Results* section (page 7, lines 188–190) to report the fold change in EC₅₀ values for each mutation. The updated sentence now reads:

"In genome-edited parasites, the TGGT1_265330 point mutations G54D, H129D, and T133M conferred reduced susceptibility to LY2090314 compared to wild-type parasites, with EC₅₀ values increasing by approximately 2.9-fold, 4.5-fold, and 4.8-fold, respectively (Figure 3B–E)."

Line 156: “Collectively, these data suggest that 157 LY2090314 directly interferes with TgGSK3 activity, resulting in impaired parasite growth”. This is a bit of an over-statement –it cannot be excluded that the mutations in TgGSK3 may confer resistance through modulating its activity, even if the target is a different protein (although I agree that this is unlikely, in view of the fact that LY2090314 is a human GSK3 inhibitor, and of subsequent structural and enzyme inhibition data presented in the manuscript). Are there any data on resistance of human cells to the inhibitor? If so, it would be of great interest if that would map to the same (conserved) residues.

We thank the reviewer for this thoughtful comment. We fully agree that, at this stage in the results description of the manuscript, indirect or more complex effects of the *TgGSK3* mutations in combination with LY2090314 treatment cannot be entirely ruled out. In response, we have revised the manuscript text (page 7, lines 194–195) to temper the original conclusion and now more cautiously state that LY2090314 likely interferes with *TgGSK3* activity.

To our knowledge, no specific LY2090314-resistant mutations in human GSK3 have been reported to date. Although certain solid tumor cell lines exhibit markedly reduced sensitivity to LY2090314 ($IC_{50} >10 \mu M$), suggesting intrinsic resistance, the mutational status of GSK3 in these lines has not been reported (PMID: 25915038).

Line 290” “promising broad-spectrum potential against apicomplexan pathogens, including *Cryptosporidium parvum*”. It would enhance the scope of this paper if the compound was shown to have activity against *Plasmodium falciparum* and *Plasmodium vivax* (in view of the need to develop antimalarials that are effective against both species, it would be useful to add *P. vivax* sequences to the phylogenetic tree in Fig. 2F).

As now mentioned in the revised version of the manuscript (Discussion, page 13), *PfGSK3* has indeed been previously explored as a potential antimalarial target. However, genetic studies indicate that *PfGSK3* is not essential during the asexual erythrocytic cycle, as parasites with disrupted *PfGSK3* remain viable, albeit with defects in gametocyte development and erythrocyte invasion (PMID: 35961464). This reduced essentiality might limit the therapeutic potential of GSK3 inhibitors against *P. falciparum*. We agree that it would be important to experimentally evaluate the efficacy of LY2090314 against *Plasmodium* species, including *P. vivax*, to fully assess its spectrum of activity across apicomplexans. However, such assays fall outside the scope of the current study, which is focused on *Toxoplasma gondii* and *Cryptosporidium parvum*. We view this as an important avenue for future investigation and would prefer to address this question in follow-up studies. Regarding the phylogenetic analysis in Figure 2F, we have now updated the tree to include *P. vivax* GSK3 ortholog in response to the reviewer’s suggestion, to provide a broader comparative framework and support the evolutionary conservation of the target. Note that the revised phylogenetic tree was generated using full-length protein sequences. The figure legend has been updated accordingly.

Line 271: “cysteine-crosslinked kinase dimers are rare, with only a handful of crystal structures in the PDB available for comparison (including, but not limited to the PDB IDs: 1ZMW, 2WNT, 5NGO, 6VPI, 6PVJ)”. It would be useful to name the kinases relating to these PDB IDs (only 6PVJ/Aurora kinase is

named). Since the authors argue below that this may apply to entire CMGC group, it would be good to indicate that these are indeed members of the CMGC group.

We thank the reviewer for this constructive suggestion. To improve clarity and strengthen the relevance of our comparison, we have revised the corresponding sentence in the Results section (page 11, lines 319–323) to name each kinase associated with the listed PDB structures. Specifically, we now indicate that the PDB entries correspond to MARK2 (1ZMW, CAMK family), ribosomal S6 kinase (2WNT, RSK family), RIP2K (5NGO, Tyrosine Kinase-Like [TKL] superfamily), and Aurora kinase A (6VPI, 6VPJ, AGC-related family). We also note that these examples fall outside the CMGC kinase group, further underscoring the unusual nature of the disulfide-linked dimer observed for *TgGSK3*.

Line 314: “This intermolecular interaction potentially represents a hallmark of CMGC kinase activation”. Could this be this an overstatement? Is this Cys residue conserved within the entire CMGC group? Many kinases of this group have been crystallised -- has any such dimerization been observed with other members of the group? Aurora kinases (which are referred to as displaying a similar Cys-mediated dimerization) are not part of the CMGC group.

We thank the reviewer for this critical observation and fully agree that our initial wording may have overstated the generality of the disulfide-linked dimerization as a feature of CMGC kinase activation. In response, we have revised the manuscript text (page 13, lines 364-365) to soften this conclusion and clarify that, to our knowledge, intermolecular disulfide bonds in crystallized kinases have only been reported in a few non-CMGC family members, including MARK2 (1ZMW, CAMK family), RSK (2WNT, AGC-related family), RIP2K (5NGO, TKL family), and Aurora kinase A (6VPI, 6VPJ, AGC-related family) (see the above comment). We have also added additional examples with disulfide-linked dimers: CIPK23 from *Arabidopsis thaliana* (4CZT, CBL-interacting protein kinase family) and STK10 (4EQU, STE family). Notably, despite the functional diversity and structural divergence of these kinases, one of the Aurora kinase A structures (6VPJ) exhibits a disulfide bond involving a cysteine residue that occupies a position analogous to that observed in *TgGSK3*. However, as correctly pointed out by the reviewer, Aurora kinases do not belong to the CMGC group, and current structural databases contain no reports of similar dimeric disulfide interactions in other CMGC kinases. We have accordingly moderated our statement in the revised Discussion section.

There are a number of papers describing the development of inhibitors against the Plasmodium orthologue PfGSK3 (see for example PMID: 36166733; PMID: 33813310; PMID: 23214499). It would be good to cite these papers, and mention if any of the compounds described therein have similarity with the series that includes LY2090314.

In the revised version of the manuscript, we have now cited the suggested publications in the Discussion section (page 13, lines 372–378). As requested, we have also included a statement clarifying that the chemical scaffolds of the *PfGSK3* inhibitors described in these studies are structurally distinct from LY2090314, which is a maleimide-based kinase inhibitor originally developed to target human GSK3.